



# Unveiling the Si cycle using isotopes in an iron fertilized zone of the Southern Ocean: from mixed layer supply to export

Ivia Closset[1], Damien Cardinal[1], Mathieu Rembauville[2], François Thil[3], Stéphane Blain[2]

[1]Sorbonne Universités (UPMC, Univ Paris 06)-CNRS-IRD-MNHN, LOCEAN Laboratory, 4 place Jussieu, F-75005 Paris, France

[2]Sorbonne Universités (UPMC, Univ Paris 06)-CNRS, Laboratoire d'Océanographie Microbienne (LOMIC), Observatoire Océanologique, F-66650 Banyuls/mer, France

[3]Laboratoire des Sciences du Climat et de l'Environnement, CNRS 91190 Gif-sur-Yvette, France

*Correspondence to*: Ivia Closset (ivia.closset@locean-ipsl.upmc.fr)

**Abstract.** A massive diatom-bloom is observed annually in the surface waters of the naturally Fe-fertilized Kerguelen Plateau (Southern Ocean). In this study, silicon isotopic signatures ($\delta^{30}$Si) of silicic acid (DSi) and suspended biogenic silica (BSi) were investigated in the whole water column with an unprecedented spatial resolution in this region, during the KEOPS-2 experiment (spring 2011). We use $\delta^{30}$Si measurements to track the silicon sources that fuel the bloom, and investigate the seasonal evolution of Si biogeochemical cycle in the iron fertilized area. We compare the results from a HNLC reference station with stations characterized by different degrees of iron enrichment and bloom conditions. Dissolved and particulate $\delta^{30}$Si signatures were generally highly variable in the upper 500 m, reflecting the effect of the intense silicon utilization in spring, while they were quite homogeneous in deeper waters. The Si-isotopic and mass balance identified a unique WW Si-source for the iron-fertilized area originating from the southeastern Kerguelen Plateau and spreading northward. However, when reaching a retroflection of the Polar Front (PF), the $\delta^{30}$Si composition of WW silicic acid pool was getting progressively heavier. This would result from sequential diapycnal mixings between these initial WW and ML water masses, highlighting the strong circulation of surface waters that defined this zone. When comparing the results from the two KEOPS expeditions, the relationship between DSi depletion, BSi production and their isotopic composition appears decoupled in the iron fertilized area. This seasonal decoupling could help to explain the low apparent fractionation factor observed here in the ML at the end of summer. Taking into account these considerations, we refined the seasonal net BSi production in the ML of the iron-fertilized area to $3.0 \pm 0.3$ mol Si m$^{-2}$ y$^{-1}$, that was exclusively sustained by surface water phytoplankton populations. These insights confirm that the isotopic composition of dissolved and particulate silicon is a promising tool to improve our understanding on the Si-biogeochemical cycle since the isotopic and mass balance allows resolving the processes involved i.e. uptake, dissolution, mixing.




## 1 Introduction

The Southern Ocean plays a crucial role in the regulation of the global climate as it contributes significantly to the world's ocean primary production and represents a major carbon sink (Takahashi et al., 2009). In the Southern Ocean, diatoms, a phytoplankton group that produce an opaline cell wall called frustule (amorphous $SiO_2.nH_2O$), are responsible for more than

75 % of the annual primary production in this ocean, in particular in the area south of the Antarctic Polar Front (PF) where they form massive blooms (e.g. Brzezinski et al., 2001). These latter are often associated with high export events as diatoms could be removed from the mixed layer (ML) via aggregation, settling or grazing processes (Boyd & Trull, 2007; Buesseler 1998) and thus contribute to the extensive biogenic silica (hereafter referred to as BSi) deposition that characterizes the sediments of this region known as the "opal belt" (Tréguer & De la Rocha, 2013; Ragueneau et al., 2000).

Since diatoms preferentially take up light $^{28}$Si isotopes, their biological activity leaves a clear imprint on the isotopic composition of both silicic acid ($H_4SiO_4$, hereafter referred to as DSi) and BSi, enriching the DSi pool with heavy $^{30}$Si isotope (De la Rocha et al., 1997). While a recent in vitro study has pointed out that this preferential uptake may vary among diatom's species (Sutton et al., 2013), field studies have reported constant Si isotopic fractionation factor ($^{30}\varepsilon$) for the

Antarctic Circumpolar Current, ACC (e.g. De la Rocha et al., 2000; Cardinal et al., 2005), estimated to -1.2 ± 0.2 ‰ on average (Fripiat et al., 2011a). On the opposite, it is not clear whether BSi dissolution fractionates silicon isotopes. Two studies based on laboratory experiments provide contradictory results. Wetzel et al., (2014) performed alkaline digestion of diatom opal in sediment cores and reported no preferential release of a given isotope. Conversely, Demarest et al. (2009) placed fresh siliceous particles in undersaturated seawater and reported a fractionation factor of -0.55 ‰ associated with the

dissolution process. This fractionation is opposite to the one occurring during Si uptake (BSi dissolution preferentially releases $^{28}$Si) which would reduce the overall (or net) fractionation factor. Thus, information on the silicon isotopic composition ($\delta^{30}$Si) potentially enables to identify DSi sources, discriminate and quantify different processes such as Si-uptake, BSi dissolution or physical mixing (e.g. Fripiat et al., 2011a).

Since they have absolute requirement for silicic acid, diatoms generate the largest latitudinal gradient of DSi from replete conditions south of the PF, in the Antarctic Zone (AZ), to depleted conditions in the Sub-Antarctic Zone (SAZ) (e.g. Brzezinski et al., 2001; Sarmiento et al., 2004). This contrasts strongly with the High Nutrient Low Chlorophyll (HNLC) characteristics of the Southern Ocean which could be more specifically defined as a High Nitrate Low Silicon Low Chlorophyll area (HNLSiLC, Dugdale et al., 1995). Several studies based on in vitro and artificial Fe-enrichments showed

that Fe stress is partly responsible for the HNLSiLC status of the ACC (see review in De Baar et al., 2005; Boyd et al., 2007). In parallel, several field studies have studied the effect of natural iron delivery on the biogeochemical cycles (e.g. Blain et al., 2007; Pollard et al. 2009).





The KEOPS project consisted of two expeditions (late-summer 2005 and early-spring 2011) conducted in a naturally iron-fertilized area in the vicinity of the Kerguelen Plateau where a massive phytoplankton bloom is observed annually (Mongin et al., 2008). The first KEOPS expedition (January-February 2005) has highlighted that this bloom is sustained by iron supply originating from iron-rich deep waters through winter mixing and vertical diffusion in summer (Blain et al., 2007). The second expedition also revealed that episodic deepening of the mixed layer (ML) could also contribute to the vertical supply of Fe (Bowie et al. 2015). Silicon isotopes were studied during this first cruise and highlighted important aspects of the Si biogeochemical cycle in both the HNLC and the fertilized area (Fripiat et al., 2011a). It was suggested that the decline of the bloom was strongly controlled by silicic acid and iron co-limitations (Mosseri et al., 2008). Results from KEOPS-2 (October-November 2011) have shown that the Kerguelen bloom was characterized by a complex and heterogeneous distribution of phytoplankton communities (e.g. Lasbleiz et al., 2014), constrained by a mosaic of biogeochemical conditions (Park et al., 2014).

Isotopic variations induced by biological Si-utilization in the Southern Ocean ML can be described in a first approximation using two different models for closed and open systems. In the closed system (also referred-to as Rayleigh model) the surface ocean is considered to have a finite pool of DSi and nutrient consumption is not replenished by any external sources. In this case, the reaction progresses in a sequential mode over time, consuming substrate (here silicic acid) that was initially present in the ML, and increasing exponentially its $\delta^{30}$Si (Eq. 1). The isotopic composition of the short-term or instantaneous product (here exported BSi, Eq. 2) differs from the long-term product that accumulates (here in the ML) and finally hold the same signature than the initial substrate when everything is consumed in the system (Eq. 3):

$$\delta^{30}Si_{sub} = \delta^{30}Si_{init} - {}^{30}\varepsilon \ln(1-f) \qquad (1)$$

$$\delta^{30}Si_{inst} = \delta^{30}Si_{sub} + {}^{30}\varepsilon \qquad (2)$$

$$\delta^{30}Si_{acc} = \delta^{30}Si_{init} + {}^{30}\varepsilon \; \frac{f \times \ln f}{1-f} \qquad (3)$$

where ${}^{30}\varepsilon$ is the isotopic fractionation factor of the reaction, f the fraction of the remaining substrate and the subscripts "sub", "init", "acc" and "inst" refer to the remaining substrate, the initial substrate, the accumulated product and the instantaneous product respectively.

A distinct system is an open flow-through system (also referred to as steady-state model) where continuous supplies of substrate balance the export of product. In this model, the DSi supply equals the sum of the BSi produced and immediately exported and the residual DSi stock leaving the system. Only one product forms from substrate and both $\delta^{30}$Si$_{DSi}$ and $\delta^{30}$Si$_{BSi}$ signatures display linear changes (Eq. 4 and 5):

$$\delta^{30}Si_{sub} = \delta^{30}Si_{init} - {}^{30}\varepsilon(1-f) \qquad (4)$$

$$\delta^{30}Si_{prod} = \delta^{30}Si_{init} + {}^{30}\varepsilon \times f \qquad (5)$$





In these ideal situations, the $\Delta^{30}Si$, i.e. the difference between $\delta^{30}Si_{DSi}$ and $\delta^{30}Si_{BSi}$, is considered as constant. In the ocean, it could actually be altered by non-steady state conditions and/or other processes such as water mixing or BSi dissolution that would affect the isotopic and mass balance (e.g. Demarest et al., 2009). Moreover, it is likely that natural environments rather proceed in a mixed way between these ideal closed and open systems, with real fractionation trend lying between the 2 curves depending on the Si-uptake:Si-supply ratio occurring in the system (Fry 2006). This could be particularly the case in the Southern Ocean where large nutrient consumption was observed in regions characterized by intense mixing events that strongly deepen the ML and bring DSi in surface waters (e.g. Brzezinski et al., 2001; Nelson et al., 2001).

In this paper, we investigate the spatial and seasonal variability of silicon isotopic composition of both seawater ($\delta^{30}Si_{DSi}$) and siliceous particles ($\delta^{30}Si_{BSi}$) in deep and surface waters of the Kerguelen area during the spring period. Our specific objectives are the following:

- Describe the early spring spatial distribution of silicon isotopes in contrasting productive (and non productive) environments off Kerguelen Islands.

- Discuss the potential role of iron in regulating the Si biogeochemical cycle and isotope dynamic in the Southern Ocean surface and subsurface waters.

- Refine the net BSi production estimated by Fripiat et al. (2011a) using the appropriate spring biogeochemical conditions (DSi concentration and $\delta^{30}Si$).

- Finally, identify and track the Si sources that fuel the phytoplankton bloom above the Plateau and characterize the temporal evolution of silicon biogeochemical cycle by combining our KEOPS-2 results with KEOPS-1 data which correspond to the bloom decline.

## 2 Material and Methods

### 2.1 KEOPS-2 cruise, hydrological setting and sampling strategy

The KEOPS-2 expedition was conducted in the Kerguelen Plateau region (Indian sector of the Southern Ocean) from 10 October to 20 November 2011 (austral spring) on board of the R/V Marion Dufresne. This Plateau acts as a barrier to the ACC whose 60 % of the total flux passes north of the Kerguelen Islands, mostly associated with the SAF, while 40 % is transported across the southern part of the Plateau and forms the jet of the PF (Park et al., 2014). This strong current is then deflected to the north following the eastern escarpment of the Plateau and forms a permanent cyclonic meandering associated to strong mesoscale activity (Park et al., 2014; Zhou et al., 2014). As a consequence, the circulation a above the central Kerguelen plateau is relatively weak (Park et al., 2008b), providing good conditions for elevated primary production (Mongin et al., 2008). These particular hydrographic features generate contrasted biogeochemical and physical environments where phytoplankton communities will respond differently to iron availability. During KEOPS-2, the vertical distribution of water masses was characteristic of the Antarctic zone (AZ) in the vicinity of the PF (Park et al., 2014). Except for the first



visit at over the Plateau (A3) where they reached the surface, the remnant Winter Water (WW, generally from 100 to 400m), where capped by a homogeneous, warm and fresh mixed layer (ML or AASW) induced by seasonal stratification. Below these subsurface waters, a subsurface temperature maximum between 400 and 1400 m associated to the Upper Circumpolar Deep Waters (UCDW), followed by an oxygen minimum between 1400 and 2600 m that correspond to the Lower Circumpolar Deep Water (LCDW) in all the out-Plateau stations. The deeper Antarctic Bottom Waters (AABW) were found only at station F-L, north of the PF. The cruise consisted in two transects north to south (TNS stations) and east to west (TEW stations) aiming at documenting the spatial extension of the bloom and its coastal-offshore gradient; and 9 long-term stations devoted to process studies (Fig. 1):

- A HNLC reference station (R2) located upstream of the eastward ACC flow, south of the Kerguelen Islands.

- 2 visits at the KEOPS-1 Plateau bloom reference station (A3-1 and A3-2).

- A productive station (E-4W) located in the plume of chlorophyll north of A3 and close to the jet induced by the PF.

- An open ocean station (F-L) influenced by warmer Subantarctic Surface Waters, located north of the PF.

- 5 stations (E-1 to E-5) constituting a pseudo-Lagrangian survey and located in the area of PF retroflection characterized by strong mesoscale activity (Zhou et al., 2014) and hereafter referred to as the Meander.

## 2.2 Sample collection and preparation

Si isotopic compositions from a total of 224 seawater samples and 137 particulate samples are presented in this study (Table S2). These latter included 91 samples collected in the ML using Niskin bottles and 46 samples coming from in situ pumps (ISP). At all stations, seawater and particles were collected using a CTD (Conductivity-Temperature-Depth) rosette equipped with 12 L Niskin bottles. Seawater (approximately 5 L) was immediately filtered through polycarbonate membranes (Nuclepore, 0.4 μm) using large volume filtration units. Filtered water samples were stored in the dark in acid-cleaned polypropylene bottles and membranes were dried overnight at 50 °C and stored in polycarbonate Petri dishes at room temperature. Since particle concentration decreases with depth, deep-water particles were collected using in-situ pumps at 7 stations (see Fig. 1) by filtering 70 to 1800 L of seawater through hydrophilic polyestersulphone membranes (SUPOR, 0.8 μm). Approximately 1/8 of the SUPOR membranes were dedicated to silicon isotopic analysis and were dried overnight at 50 °C and stored in Petri dishes at room temperature.

A moored sediment trap (Technicap PPS3) was also deployed at station A3 over the central Kerguelen Plateau at 289 m (seafloor depth 527 m). The sediment trap carrousel was composed of 12 sampling cups (250 mL) collecting sinking particles from the 21 October 2011 to 7 September 2012. Sampling intervals were programmed to be short (10-14 days) in spring and summer and longer (99 days) in autumn and winter. The description of the physical environment of the deployment, together with the detailed sediment trap samples processing are reported in Rembauville et al. (2015a). After





BSi extraction (Rembauville et al., 2015b), samples were purified and analyzed for the Si isotopic composition as described below.

### 2.3 Sample collection and preparation

### 1.2.1 Particles digestion and BSi analyses

The membranes (Nuclepore and SUPOR) were subjected to a wet-alkaline digestion (adapted from Ragueneau et al., 2005). BSi was dissolved in Teflon tubes using a 0.2 µmol L$^{-1}$ NaOH solution (pH 13.3) at 100°C for 40 min followed by neutralization with HCl (1 mol L$^{-1}$). As this digestion can also solubilize some lithogenic silica (mainly clay minerals), a second and identical digestion was applied to the membranes, and aluminum (Al, a tracer of lithogenic source) was analyzed in the two leachates using an Inductively Coupled Plasma Mass Spectrometer (ICP-MS; detection limit = 3.18 ppb). Using

the Si:Al ratio measured in the second digestion, potential lithogenic silicon dissolved in the first digestion can be estimated (Ragueneau et al., 2005). Unfortunately, polycarbonate membranes used for silicon isotopic analysis were contaminated by Al during filtrations onboard. However, SUPOR filters as well as Al concentrations estimated on other filters (Lasbleiz et al., 2014) revealed negligible lithogenic silicon in the first leachate (on average 1.26 % for SUPOR filters). Such a lithogenic contribution should not bias significantly our $\delta^{30}$Si value. Indeed, using a light end-member $\delta^{30}$Si$_{LSi}$ of -2.3 ‰ reported in

clays (Opfergelt & Delmelle, 2012) and our maximum $\delta^{30}$Si$_{BSi}$ (2.06 ‰ station R2, 2400m) as extreme end-members, we calculated a maximum interference in the isotopic signal of 0.05 ‰ which is similar to our analytical precision for $\delta^{30}$Si. BSi concentrations were determined with a colorimetric method according to Grasshoff et al. (1999) and by ICP-MS on the same samples as for Si-isotopic composition. Every ISP samples has been analyzed in full duplicates (i.e. on a second fraction of the same membrane) with a pooled standard deviation of 5.0 ± 4.6 % (n = 58), which corresponds to our average

reproducibility of BSi measurements, and which is slightly better than the uncertainty estimated for this method (10 %, Ragueneau et al., 2005). Moreover, independent measurements of BSi concentration performed with Niskin bottles at the same stations and depths during KEOPS-2 by Lasbleiz et al. (2014) were similar to our results, suggesting the ISP method was robust.

### 1.2.2 Seawater preconcentration and DSi analyses

A two-step preconcentration procedure adapted from the MAGIC method (Karl & Tien, 1992; Reynolds et al., 2006) was performed on seawater samples to increase H$_4$SiO$_4$ concentration and reduce the anionic matrix that could interfere with Si during isotopic analysis (e.g. sulfates, SO$_4^{2-}$; Hughes et al., 2011). DSi was co-precipitated in two steps with brucite (Mg(OH)$_2$) by adding 2 % (v/v), following by 1 % (v/v) of 1M NaOH to the seawater sample. This solution was shaken and

left for 1h and the precipitate was recovered by centrifugation and redissolved with 1M HCl. The supernatant was removed and complete Si recovery was monitored by checking systematically that no detectable amount of silicic acid remained in the




supernatant after coprecipitation and centrifugation. DSi concentrations in seawater samples were determined with a colorimetric method (Grasshoff et al., 1999) on the same samples as for Si-isotopic composition. Average reproducibility of DSi measurements was 6.7 % (calculated from 98 in-house silicon solution analyses at the ± 1 sd level).

## 2.4 Purification

Separation of Si from other ions in the sample was achieved by passing the solution through a cation-exchange column (BioRad cation exchange resin DOWEX 50W-X12, 200 to 400 mesh, in H+ form) using a protocol described in Georg et al. (2006). After purification systematic analysis of major elements (such as Mg, Ca, Na, Al) was performed by ICP-MS to ensure the sample purity prior to isotopic analyses (Si/X weight ratio usually > 50). Si concentrations were also measured in the purified solutions to check for complete recovery. This purification step did not allow the complete removal of the anionic matrix, which consists primarily of $Cl^-$, $SO_4^{2-}$ and to a lesser extent $PO_4^{3-}$ and $NO_3^-$. $Cl^-$ originating from seawater can be neglected compared to $Cl^-$ added as HCl to dissolve the brucite. Therefore the solutions dedicated to DSi isotopic measurements were analyzed by anionic chromatography to control the concentration of sulfates. Indeed, in these samples, $SO_4^{2-}$ concentrations could induce a significant shift in isotopic measurements (see supplementary method and Van den Boorn et al. (2009) for rock digestion solutions). Thus, as proposed by Hughes et al. (2011), samples and standards used for DSi isotopic analyses were doped with sulfuric acid in large excess compared to the natural $SO_4^{2-}$ concentrations in order to control this sulfate matrix effect.

## 2.5 Isotopic measurements

The purified and sulfates doped Si solutions were analyzed for isotopic measurements on a Thermo Neptune$^+$ Multicollector Inductively Coupled Plasma Mass Spectrometer (MC-ICP-MS; LSCE, Gif-sur-Yvette) in dry plasma mode using Mg external doping to correct for the mass bias (Cardinal et al., 2003; Abraham et at., 2008). Samples were injected into the plasma with an Apex desolvating nebulization system connected with a PFA nebulizer (100 μL min$^{-1}$ uptake rate) and without additional gaz. Silicon isotopic compositions ($\delta^{30}$Si) were calculated as the permil deviation from the quartz standard NBS28 (RM8546). They were measured relative to an in-house standard Quartz Merck, which was not significantly different from NBS28 (Abraham et al., 2008), analyzed immediately after and before the sample and expressed as:

$$\delta^{30}Si\ (‰) = \frac{^{30}Si\ ^{28}Si\ sample}{^{30}Si\ ^{28}Si\ standard} - 1\ \times 1000 \qquad (6)$$

Blanks levels were below 1 % of the main signal and were subtracted from each sample and standard analyses. All measurements were carried out in a matrix composed of $HNO_3$ 0.5 mol L$^{-1}$, HCl 0.5 mol L$^{-1}$, $H_2SO_4$ 1 mmol L$^{-1}$ and medium





resolution mode (M/ΔM > 6000) to optimize the separation of $^{30}$Si peak and $^{14}$N$^{16}$O interference and were performed on the interference free left side of the peak (Abraham et al., 2008). $\delta^{29}$Si and $\delta^{30}$Si were compared to the mass dependent fractionation line (Fig. S3) and samples falling outside of its analytical error were excluded from final dataset. Typical analytical conditions are provided in table S3.

Numerous analyses of a secondary reference material such as Diatomite ($\delta^{30}$Si = 1.26 ‰, Reynolds et al., 2007) generated over the entire procedure indicated an average precision and a long-term analytical reproducibility (24 months) on $\delta^{30}$Si values of 1.28 ± 0.05 ‰ (1 sd, n = 128) and confirmed that no uncorrected isotopic bias occurred. All BSi samples and some DSi samples have been fully replicated and measured on separate days (chemical preparation plus isotopic measurements).

In these cases, the average global reproducibility on full duplicates $\delta^{30}$Si is 0.06 ‰ (1 sd, n= 108) and 0.04 ‰ (n= 78) for $\delta^{30}$Si$_{BSi}$ and $\delta^{30}$Si$_{DSi}$ respectively. Error bars shown in all figures correspond to the analytical reproducibility or to the global reproducibility if greater than 0.05 ‰.

## 3 Results and Discussion

### 3.1 General considerations

During KEOPS-2, $\delta^{30}$Si$_{DSi}$ displayed a clear inverse relationship with silicic acid concentration as commonly observed in the Southern Ocean (Fig. S4 and e.g. in Fripiat et al., 2012; De Souza et al. 2012). The water column profiles showed a general increase in DSi concentrations. $\delta^{30}$Si$_{DSi}$ became gradually lighter with depth. In the upper 500 m, $\delta^{30}$Si$_{BSi}$ values were systematically lighter than $\delta^{30}$Si$_{DSi}$ values in agreement with the preferential uptake of $^{28}$Si by diatoms (De La Rocha et al.,

1997). Below 500 m there was still a slight increase of $\delta^{30}$Si$_{BSi}$ and a decrease of $\delta^{30}$Si$_{DSi}$ values coincident with increasing DSi concentrations. In contrast to the ML, $\delta^{30}$Si$_{BSi}$ were there systematically higher than $\delta^{30}$Si$_{DSi}$ (on average 1.74 ± 0.13 and 1.20 ± 0.17 ‰, respectively). This observation differs significantly from the one observed in the Atlantic sector in summer by Fripiat et al. (2012) where the isotopic signature of BSi exported to depth was directly comparable to the $\delta^{30}$Si$_{BSi}$ values of particles in the ML. The surface KEOPS-2 $\delta^{30}$Si$_{BSi}$ (ranging from 0.47 ‰ to 2.04 ‰) and $\delta^{30}$Si$_{DSi}$ (ranging from 1.96 ‰ to

2.79 ‰) encompass nearly the full range of delta results reported from previous Southern Ocean studies (-0.7 ‰ to 2.8 ‰ for BSi and 0.5 ‰ to 4.4 ‰ for DSi, e.g. Cao et al., 2012; De Souza et al., 2012; Fripiat et al., 2012).

In general, $\delta^{30}$Si$_{DSi}$ signatures in deep waters were homogeneous and did not show large changes with depth or between stations (on average 1.28 ± 0.08 ‰ and 1.05 ± 0.06 ‰ for UCDW and LCDW respectively). There is a much higher

variability in $\delta^{30}$Si$_{DSi}$ in the ML and WW between stations (1.99 ‰ to 2.53 ‰ and 1.46 ‰ to 2.03 ‰, respectively, Table S4). Note that the WW exhibited systematically the largest dispersion of delta values and were systematically associated to a strong isotopic gradient toward light $\delta^{30}$Si$_{DSi}$ with depth (not shown). Consequently, the mean and median values were





probably not representative of the real WW Si-properties. Usually, the temperature-minimum layer (T-min between 1.5 and 2°C in the Kerguelen area; Park et al., 2014) is chosen as the most traditional definition of the remnant surface winter water. However, in some stations, it was not possible to clearly determine T-min (e.g. R2 and E2) and/or there was a significant salinity gradient above the T-min depth (e.g. TNS01, F-L). Trull et al. (2015) have proposed that a shallower depth based on a threshold increase in salinity of 0.05 (S-threshold depth) could better represent the WW characteristics. They ascribed this situation to a weaker winter mixing compared to previous year and thus, the nutrient depletion between the T-min and S-threshold depths could not be associated to recent consumption. In the following we will use the S-threshold approach since it would be the most appropriate to reflect the pre-bloom biogeochemical conditions.

## 3.2 Distribution of Si isotopes vs. source and supply of iron

### 3.2.1 The HNLC reference station

In the ML, the productive and HNLC areas exhibited significant different silicon isotopic composition of both seawater and particles, reflecting the different degrees of Si-utilization by diatoms. The HNLC reference station (R2) displayed low chlorophyll a and BSi and very low Si-uptake rates, consistent with its iron-depleted condition and the dominance of non-siliceous organisms (see details in Lasbleiz et al., 2014 and Closset et al., 2014). In surface waters, the concentration of biogenic silica was the lowest measured in surface during KEOPS-2 ($0.30 \pm 0.03$ µmol L$^{-1}$) and its silicon isotopic composition was low ($0.73 \pm 0.04$ ‰) and similar to E4-W and A3-1, Fig. 2e, 2f). This is typical of non-bloom conditions and in the same range as those measured in HNLC waters of the Southern Ocean (e.g. Fripiat et al., 2011a, 2011b; Mosseri et al., 2008). However, the HNLC station displayed unexpected low silicic acid concentration and heavy $\delta^{30}Si_{DSi}$ in the ML ($12.94 \pm 0.49$ µmol L$^{-1}$ and $2.21 \pm 0.06$ ‰, respectively; Fig. 2a, 2b). This latter is significantly heavier than the Si-isotopic composition of the fertilized area measured few days before ($1.99 \pm 0.03$ ‰, A3-1). As already proposed by Closset et al. (2014) and Lasbleiz et al. (2014), this suggests that a development of diatoms could have occurred before our sampling, consuming a fraction of the DSi standing stock and increasing the $\delta^{30}Si_{DSi}$ of surface waters. The same evidence of surface production has also been deduced by Dehairs et al. (2015), who observed a slight nitrate depletion and enrichment of $\delta^{15}N$-NO$_3^-$. Indeed, in early spring, the low iron concentration that prevails at this station might be sufficient to trigger a short phytoplankton growth (dominated by nanophytoplankton; Lasbleiz et al., 2014) as soon as light conditions became favorable. Then, both the high silica dissolution to production ratio (D:P > 1) observed in the ML (Closset et al., 2014) and the high barium excess measured between 200 and 400 m (Jacquet et al., 2015) suggest that this material was exported and remineralized when we visited the station. This could be confirmed by the clear $\delta^{30}Si_{BSi}$ maximum observed between 100 and 200 m (Fig. 2f) that could result from the dissolution isotopically light diatoms that were initially produced from lighter DSi.




By contrast, all other KEOPS-2 stations were characterized by the development of large spring blooms that were not homogeneous in time and space depending on the degree, the mode and the timing of their iron fertilization (Bowie et al., 2015; Trull et al., 2015). These blooms were organized in three main clusters related to their different iron supplies and are discussed separately in the following.

### 3.2.2 The Kerguelen Plateau zone

This area was characterized by large and recurrent blooms located southeast of the islands, mainly above the Kerguelen Plateau and delimited northward by the Polar Front (Blain et al., 2001, 2007). During our first visit to A3 (A3-1), low $\delta^{30}Si_{DSi}$ and $\delta^{30}Si_{BSi}$ (1.99 ± 0.03 ‰ and 0.77 ± 0.05 ‰, respectively, Fig. 2b, 2f) were measured in the ML indicating that

biogeochemical conditions prevailing there were characteristics of a pre-bloom or early-bloom period. Indeed, low chlorophyll a and BSi concentrations were observed despite high nutrients standing stocks ($H_4SiO_4$ and $NO_3^-$, see Blain et al., 2015 for $NO_3^-$) and relatively high iron concentrations (Bowie et al., 2015).

The largest phytoplankton development was observed during the second visit at A3, where chlorophyll a and BSi

concentrations increased more than two fold over one month. This growth was reflected by the significant increase of both $\delta^{30}Si_{DSi}$ and $\delta^{30}Si_{BSi}$ (from 1.99 ± 0.03 ‰ to 2.10 ± 0.05 ‰ and from 0.77 ± 0.05 ‰ to 0.96 ± 0.08, respectively, Fig. 2b, 2f). There, the supply of nutrients and iron to the ML coming from the WW both in winter and during the productive period allowed a spring biogenic silica production comparable to the highest productive regions, such as upwelling systems or river plumes (up to 43.4 ± 0.4 mmol m$^{-2}$ d$^{-1}$ for net opal-production integrated over the euphotic zone, Fripiat et al., 2011a; Closset

et al., 2014).

### 3.2.3 The recirculation zone in the Polar Front meander

The central part of the meander was characterized by a complex and slowly flowing water circulation associated with low to moderate dissolved iron concentrations (Bowie et al., 2015; Quéroué et al., 2015). Here, surface waters displayed generally

higher $\delta^{30}Si_{DSi}$ signatures and lower DSi concentrations compared to above the Plateau (TNS6, E1 to E5; Fig. 2d), supporting the idea of a northward surface circulation with a progressive consumption and enrichment in $^{30}Si$ of the dissolved pool. According to Park et al. (2014), this zone could correspond to the latest arrival of water originated from the shallow Plateau located South of these stations. This would explain the delay observed between the initiation of production here and the southernmost bloom located above the Plateau and in the PF plume (Fig. 1). Moreover, radium isotopes signature (Sanial et

al., 2015) suggested that the southward transport of chemical elements (such as iron) across the PF could also occur and significantly fuel the phytoplankton bloom in this area, mixing the Si-poor PFASW with heavy $\delta^{30}Si_{DSi}$ and Si-rich AASW with light $\delta^{30}Si_{DSi}$.



The moderate iron fertilization occurring in the meander should have increased both the BSi production rate and biomass, with larger increase at the two last visits (E4E and E5, Closset et al., 2014; Lasbleiz et al. 2014). However, no clear trend was identified in the ML values of $\delta^{30}Si_{DSi}$ and $\delta^{30}Si_{BSi}$ since these two parameters were not significantly different from

TNS6 to E4E (Table S4). This observation is strengthened by the homogeneous $\delta^{30}Si_{DSi}$ profiles in all the meander's stations (Fig. 2d). TSN6 and E1 to E3 stations display relatively homogeneous BSi concentration and isotopic composition. E4E and E5 are characterized by an increase in sub-surface for these two parameters. Indeed, these two stations were representative of "spring-bloom stations" with high net silica production rates and low dissolution to production ratios (Closset et al., 2014) and thus displayed a high potential for BSi accumulation in the ML. Interestingly, the $\delta^{30}Si_{BSi}$ profiles suggest that this

accumulation was confined in the lower part or just below the ML since we observe a significant $\delta^{30}Si_{BSi}$ maximum between 50-150 m (up to 1.46 ‰, Fig. 2h). Such a subsurface increase of diatom's $\delta^{30}Si$ signature could result from a significant BSi accumulation in the deeper layers. Lasbleiz et al. (2014), associated the low biomass in the meander surface layer to the occurrence of a deep silica maximum, and suggested that the circulation pattern in this area may have favored the transfer and the accumulation of particles at depth. Indeed, using [234]Th, Planchon et al. (2015) measured the highest export

efficiencies at stations E but suggested that relatively low carbon export occurred inside the meander during the onset of the bloom. Thus, the heavy Si-isotopic composition of particles observed at depth at station E4E and E5 might result from the early exported surface layer diatoms that would accumulate at the base of the ML, as already proposed in a conceptual scheme by Quéguiner (2013).

## 3.2.4 The Polar Front Zone

A plume of chlorophyll a was present in the Polar Front Zone (stations F-L and TEW8), extending eastward of the Kerguelen Plateau and showing strong mesoscale activity associated to the temporal and spatial variability of the Polar Front (Park et al., 2014). There, the productive waters were characterized by a shallow and relatively warm ML (> 4°C) associated to moderate to high iron concentration (Quéroué et al., 2015; Bowie et al., 2015) and providing favorable conditions for

phytoplankton development. These waters were characterized by phytoplankton communities distinct than those found above the Plateau (mostly small diatoms and nanoflagellates, Lasbleiz et al., 2014). The Si-isotopic composition of ML seawater and particles measured at F-L were the heaviest of KEOPS-2 stations coinciding with a strong Si-depletion in surface water (Fig. 2b, 2f). These $\delta^{30}Si_{DSi}$ signatures fall in the range of previous reported values in the PFZ ML, from 2.05 ± 0.03 ‰ in spring (Cardinal et al., 2005) to 2.77 ± 0.23 ‰ in late summer (Fripiat et al., 2011a) in the Australian and Atlantic

sector of the Southern Ocean respectively. In the PFZ, heavier $\delta^{30}Si$ signatures can be attributed to a higher utilization of "new" silicic acid by diatoms leading to high net BSi production (Closset et al., 2014). Note that higher $\delta^{30}Si_{DSi}$ values were also measured in the PFZ WW (on average 2.14 ± 0.08 ‰ for all stations located north of the PF, Fig. 2b). This confirms



that Si source is different in the PFZ when compared to the AZ. These PFZ waters can be advected from coastal water from the northern Kerguelen shelf (Bowie et al., 2015; Park et al., 2014).

### 3.3 Spatio-temporal variability of Si mass and isotopic balance in the iron fertilized area

By comparing the present dataset with with the one collected during KEOPS-1 (summer 2005) the seasonality of the Si isotopic composition can be described, helping to refine the Si biogeochemical cycle in this area (Fripiat et al., 2011a; De Brauwere et al., 2012). To this purpose, the short term temporal evolution of the bloom initiation (two visits at A3) was completed with the several visits in the meander that constituted a pseudo-lagrangian time-series (covering 27 days). The silicon isotopic properties measured in the ML during the austral spring (KEOPS-2) were significantly lighter than those measured by Fripiat et al. (2011a) at the end of summer (Fig. 3) as expected from the seasonal progression of the preferential light Si isotopes consumption (De la Rocha et al., 1997). Temporally, the bloom above the plateau usually peaks in late November and declines gradually until January as nutrients (mainly iron and silicic acid) became limiting (Blain et al., 2007; Mongin et al., 2008). Then, a second and less important bloom could persist at steady state until May when light level started to be insufficient to maintain photosynthetic activity (Blain et al., 2013). The first visit to A3 revealed relatively high DSi concentrations and low BSi and chlorophyll *a* stocks in the ML and WW that can be associated to the initial conditions prevailing before the summer stratification (Park et al., 2014; Blain et al. 2015) whereas concentrations and $\delta^{30}$Si measured at A3 during KEOPS-1 represent the conditions at the end of the season (Fig. 3). These initial conditions (31.55 ± 2.21 µmol L$^{-1}$ and 1.76 ± 0.03 ‰ for A3-1 WW DSi concentration and isotopic composition respectively) differ from the HNLC WW chosen by Fripiat et al. (2011a) as ultimate Si-source fuelling the bloom above the Plateau (52.5 ± 3.3 µmol L$^{-1}$ and 1.5 ± 0.0 ‰ for DSi concentration and $\delta^{30}$Si$_{DSi}$, respectively), but were not significantly different from the A3 WW reported by Fripiat et al. (2011a). Indeed, ML $\delta^{30}$Si$_{DSi}$ in the early spring were clearly off the steady state fractionation trend (not shown) when using the same initial Si-pool conditions as in Fripiat et al. (2011a). Thus, it appears that the WW Si-pool isotopic and contents properties of HNLC KEOPS-1 or KEOPS-2 reference stations could not be ascribed as a common Si-source. In the meander, the initial conditions were represented by the Si-properties of TNS6 WW (31.3 µmol L$^{-1}$ and 1.71 ± 0.02 ‰ for DSi concentration and $\delta^{30}$Si$_{DSi}$, respectively) which were similar to those used for the Plateau. This strengthens the idea of a unique Si-source originated from the south and that flow northward above the shallow Kerguelen Plateau to reach finally the PF retroflection area. Thus, using the mean ACC $^{30}\varepsilon$ value of -1.2 ± 0.2 ‰ compiled by Fripiat et al. (2011a) and the averaged Plateau-WW as initial conditions, we will attempt to describe the seasonal dynamic of the Si-biogeochemical cycle in the fertilized area off Kerguelen Islands.





### 3.3.1 Mixed layer dissolved Si-pool

Using KEOPS-1 HNLC WW characteristics ($52.5 \pm 3.3$ µmol Si L$^{-1}$ and $1.5 \pm 0.0$ ‰) to represent the initial conditions of WW in the fertilized area, Fripiat et al. (2011a) estimated a seasonal depletion in the ML at $5.0 \pm 0.3$ mol Si m$^{-2}$ y$^{-1}$. By identifying a more appropriate Si-source for the Plateau ML, we can refine this calculation and reduce the seasonal net BSi

production in the upper 100 m to $3.0 \pm 0.3$ mol Si m$^{-2}$ y$^{-1}$. This flux is 40 % lower than the previous estimate but corresponds well to the range of published net BSi production values for the AZ (2.4 to 3.3 mol Si m$^{-2}$ y$^{-1}$; Pondaven et al., 2000; Nelson et al., 2002; Pollard et al., 2006) and still balance the total Si supply estimated for the Plateau ML ($4.0 \pm 0.7$ mol Si m$^{-2}$ y$^{-1}$, Fripiat et al. (2011a). The mean net BSi production occurring during the 27 days that separate the two samplings at A3 can be also estimated at 10.7 mmol m$^{-2}$ d$^{-1}$ using the steady state equations and is well consistent with the 14.3 mmol m$^{-2}$ d$^{-1}$

computed for the same period by Closset et al. (2014) in their seasonal budget of Si cycle above the Plateau.

The seasonal evolution of $\delta^{30}Si_{DSi}$ in the fertilized area seems to be better described by steady state fractionation equations (Fig. 4). This model appears appropriate to describe the Si-utilization that occurred above the Kerguelen Plateau where deep and regular mixing events are expected to supply nutrients in surface waters (Park et al., 2008b). Indeed, in Fig. 4, most of

the Plateau and Meander ML $\delta^{30}Si_{DSi}$ stations were clearly off the Rayleigh fractionation trend. It seems that ML aligns along a steady state trend with the decreasing f (i.e. increasing Si utilization): A3-1, A3-2, E1 to E4E Meander stations, E5 and A3 KEOPS-1. Surprisingly, in this model, Meander stations exhibit higher isotopic signatures and higher DSi depletion compared to the productive Plateau stations A3-2 and E4W while they were sampled several days before these two stations. This situation was not due to a higher Si-uptake in the Meander ML as the net BSi production at A3-2 was twice the one

measured at E5 (Closset et al., 2014), but could be explained by a strong mixing event that occurred just before our second visit to A3. This vertical mixing was induced by strong temporary winds and supplied water with high silicic acid content and light $\delta^{30}Si$ in the euphotic zone. This would thus decrease the apparent Si utilization (f) and lighten the isotopic signature of seawater. Note that, as previously mentioned, no clear temporal evolution can be shown in the Meander ML $\delta^{30}Si_{DSi}$ signatures. The hydrological conditions that lead to ML instabilities and regular mixing of the water masses was likely the

reason of this situation as we will discuss in the following.

### 3.3.2 Winter Waters dissolved Si-pool

By comparing WW DSi-properties from KEOPS-1 HNLC and fertilized area ($52.5 \pm 3.3$ µmol Si L$^{-1}$ and $34.2 \pm 1.9$ µmol Si L$^{-1}$), Fripiat et al. (2011a) estimated a seasonal depletion in the 100 to 400 m layer of $5.5 \pm 0.3$ mol Si m$^{-2}$ y$^{-1}$ and ascribe it to

a net BSi production of a subsurface diatom population since a deep BSi and chlorophyll maximum was observed here in January (Mosseri et al., 2008; Uitz et al., 2009). The WW characteristics measured during our first visit at A3 ($31.5 \pm 2.2$ µmol Si L$^{-1}$ ; $1.76 \pm 0.03$ ‰) were similar to the one observed at the end of the productive season by Fripiat et al. (2011a)





suggesting that, contrary to what was previously proposed, WW did not have undergone seasonal Si uptake above the Plateau. This conclusion is not contradictory with the development of a subsurface diatom community that could partly explain the deep BSi accumulation observed here during KEOPS-1. Indeed, in such communities, the BSi production may likely be sustained by regenerated source of Si (high D:P ratio, Closset et al., 2014) or may benefit from high diffusive fluxes

($> 1$ mmol m$^{-2}$ d$^{-1}$, Rembauville et al., 2016) that characterize these transition layers and would not finally consume the DSi standing stock.

In contrast to the ML, the WW $\delta^{30}Si_{DSi}$ signatures of the fertilized area (Table S4) display significant variations (from 1.71 ± 0.02 ‰ at TNS6 to 2.21 ± 0.02 ‰ at E4E). It seems unlikely that such shift of WW toward higher $\delta^{30}Si$ values are explained

by the progressive consumption of silicic acid by a secondary diatom community just below the ML. Indeed, although deep silica maximum are common features in the Southern Ocean (Parslow et al., 2001), we did not measure any Si uptake below the euphotic layer during KEOPS-2 (Closset et al., 2014). Since this area hold strong mesoscale activity (Zhou et al., 2014; Park et al., 2014), an alternative process that could have decreased the WW DSi pool and enriched the WW $\delta^{30}Si_{DSi}$ may likely be the diapycnal mixing between initial WW and several sequential surface ML water masses (Fig. 5). In this case, the

resulting water mass should lie on a theoretical mixing curve involving the TNS6 WW and different ML water masses with Si-properties located between the initial and final ML end-members (TNS6 ML and E5 ML). For example E1 WW could be fully explained by the mixing between TNS6 WW and TNS6 ML, E2 WW by the mixing between TNS6 WW and E1 ML etc. Although this hypothesis seems valuable for these two examples, the large variability of the Si isotopic properties of WW water masses prevent us to identify with sufficient precision the exact end-members of these mixings. Consequently,

only a range of potential mixing curves has been identified in Fig. 5, but all the Meander WW fall between these two extreme situations. Thus, when the water masses joined the meander, mixings between WW and ML lead to an enrichment of the Si isotopic signature and a dilution of the DSi pool of this source.

### 3.3.3 Mixed layer particulate Si-pool

While the open model equations seem appropriate to describe the evolution of DSi isotopic composition, it is arguable when considering the $\delta^{30}Si_{BSi}$, since this model conceptually assumes the lack of BSi accumulation (Fry, 2009). As shown in Fig. 4, the signatures of opal during the two KEOPS expeditions were clearly off the steady-state fractionation trend for BSi and lied in between the two product-curves involved in the closed-model. Indeed, the Rayleigh fractionation equations describe two extreme ideal situations: all the BSi produced in the ML is immediately exported from the system (the so-called

instantaneous BSi, Eq. 2), or all BSi accumulates in the ML (the so-called accumulated BSi, Eq. 3). During diatom development, and all along the season in the Kerguelen fertilized area, the system shifted from one situation to another or remained between these two extremes depending on the bloom maturity and/or the hydrodynamical conditions of the surface waters. The instabilities of the ML depth at TNS6, "bloom-initiation" stations (E1, E2 and E3) and at E4E impacted the





phytoplankton growth as discussed previously and may have led to important biomass export through detrainment. In these systems, diatoms did not accumulate in the ML but settled toward deeper layers, exporting carbon and BSi (Laurenceau-Cornec et al., 2015; Planchon et al., 2015). Their Si-isotopic properties felt close to the instantaneous product curve of the Rayleigh fractionation law (Fig. 4). Considering the early stage of the season during KEOPS-2 expedition, and since Si-

uptake rates above the Plateau were among the highest reported so far for the Southern Ocean, it appears that BSi and C export was very low at A3 and E4W (Jacquet et al., 2015; Planchon et al., 2015; Rembauville et al., 2015a, 2015b). There, BSi was accumulating in the ML, resulting in Si-isotopic properties that seemed to be better described by the accumulated product curve of the Rayleigh model. The last two stations (E5, KEOPS-2, and A3, KEOPS-1) lied between these two ideal curves since they combined the two biogeochemical processes (BSi accumulation and export). Indeed, the high Si-uptake

rates observed at E5 ($20.5 \pm 0.2$ mmol m$^{-2}$ d$^{-1}$; Closset et al., 2014) associated to the strengthening of the summer stratification would progressively reduce vertical Si supply and allow a better phytoplankton retention in the ML. KEOPS-1 expedition occurred during the decaying phase of the bloom. Thus, the biological material present in the ML at the end of summer was likely composed of old and detrital diatoms that was partly exported in December (Rembauville et al., 2015b) and living cells that were just produced from regenerated Si-sources (Closset et al., 2014). The $\delta^{30}Si_{BSi}$ observed in these

surface waters would result from the combination of a partial export of old particles which were remaining in the ML, lightening its isotopic signature, and of new and isotopically heavier diatoms that would increase the delta value. This situation is consistent with the strong seasonality of primary and export production that characterized the Southern Ocean and that leads to a temporal decoupling between these two processes in the ML (see e.g. in Rembauville et al., 2015a).

The contrasted seasonal evolutions of $\delta^{30}Si_{DSi}$ and $\delta^{30}Si_{BSi}$ (following an open or a closed system respectively) is similar to BSi and DSi isotopic offsets previously reported (e.g. Varela et al. 2004; Fripiat et al., 2012) leading to spatio-temporal variations of the ML $\Delta^{30}Si$ ($\Delta^{30}Si = \delta^{30}Si_{BSi} - \delta^{30}Si_{DSi}$), with low $\Delta^{30}Si$ at the end of summer when the ML silicic acid pool was highly depleted. Fripiat et al. (2012) attributed such offsets to modifications of the Si-uptake to Si-supply ratio in surface waters due to spatial variability of the biogeochemical conditions among the different zones of the ACC (AZ vs. PFZ). They

suggested that when the Si-uptake to supply ratio was low, (i.e. limited diatom growth and significant vertical mixing) the ML was supplied with isotopically light DSi, lightening the $\delta^{30}Si_{DSi}$ signature of surface waters without impacting the isotopic composition of biogenic silica. This process would result in a low $\Delta^{30}Si$. Our results point out that an alternative approach to change the $\Delta^{30}Si$ could be obtained when the system behaves in way that combine the open and closed systems with a decoupling between the dissolved and particulate pools. In this case, opal accumulation in the ML will increase the

$\Delta^{30}Si$ while when biogenic silica is exported in deeper layer, the $\Delta^{30}Si$ will be dampened and will differ significantly from $^{30}\varepsilon$. The export of BSi outside of a closed system is also the only situation that could explain $\delta^{30}Si_{BSi}$ values exceeding Si-source as observed at the end of summer above the Plateau (Fig. 4). In this case, diatoms remaining in the ML would represent an instantaneous product strongly enriched in heavy Si-isotope while when surface waters are strongly stratified; they could be retained in the ML and would correspond to an accumulated product.



### 3.3.4 Deep particulate Si fluxes

BSi export flux collected in the sediment trap located above the Plateau shows two summer maxima (> 2 mmol m$^{-2}$ d$^{-1}$ in early December 2011 and early January 2012) separated by a period of reduced fluxes (Fig. 6). In contrast, export during austral winter exhibited a long period of very low particle flux (<0.2 mmol m$^{-2}$ d$^{-1}$ from March to September 2012). These seasonal variations can be associated to the evolution of surface chlorophyll a concentrations with a delay of approximately one month, and were described in details in Rembauville et al. (2015a). This dataset allows us to investigate the variations of exported opal $\delta^{30}$Si and subsequently the mechanisms governing the seasonal variations of Si stocks and fluxes in the ML. Indeed, recent studies have suggested that the seasonality of silicon isotopic composition of exported BSi may reflect the evolution of $\delta^{30}$Si$_{BSi}$ in the ML (Varela et al., 2004; Closset et al., 2015). In Fig. 6, we can identify 5 key periods in the seasonal Si biogeochemical cycle above the Kerguelen Plateau:

(i) Early spring: The relatively low $\delta^{30}$Si (1.34 ‰ in October to 1.23 ‰ in December) cannot be directly related to the isotopic signature of diatoms in the ML measured at the same station for the same period (0.79 ± 0.1 ‰ late October, and 1.00 ± 0.09 ‰ mid-November) but correspond to the signature of particles collected between 200 m and 450 m (1.21 ± 0.1 ‰ late October, A3-1, and 1.34 ± 0.07 ‰ mid-November, A3-2; Fig. 2f). Although Demarest et al. (2009) have estimated a fractionation factor of -0.55 ‰ during the dissolution of BSi, it seems unlikely that this process would be responsible to this heavy isotopic signature since dissolution rates must be low during early spring (the dissolution to production ratio measured in the ML during KEOPS-2 was very low above the Plateau, less than 0.1 Closset et al., 2014). Moreover, recent studies have pointed out that the isotopic signature of particles is well conserved through the water column and can be closely related to the DSi consumption in surface waters (Fripiat et al., 2012; Closset et al., 2015). Therefore, considering that the $\delta^{30}$Si of exported BSi follow either accumulated Rayleigh or a steady state equations, the 1.23 ‰ signature measured during the first export event corresponds to a DSi consumption ranging between 45 % and 60 % of the winter DSi standing stock in the ML, respectively and results to a DSi concentration remaining in the ML ranging between 18 μmol L$^{-1}$ (Rayleigh) and 13 μmol L$^{-1}$ (steady-state) in November 2011 (one month delay between the signal produced ML and recorded in the trap). The concentration calculated from Rayleigh accumulated product equation is consistent with the DSi concentration measured in the ML during the second visit to A3 in November (19.7 ± 0.3 μmol L$^{-1}$, Fig. 2a) and confirms that, in early spring, the BSi production in the ML follows more likely a closed mode rather than an open mode.

(ii) Early summer: During December, the δ30Si of settling particles increased from 1.23 to 1.99 ‰ as a result of diatoms activity in the ML. Applying the Rayleigh equations, we can estimate a consumption of 71 % of the winter DSi standing stock. The remaining DSi concentration in the ML would be 9.53 μmol L$^{-1}$ in November 2011 which is close to 12 μmol L$^{-1}$ measured mid-November at E5. Such increase between the two BSi delta values corresponds to a net BSi production of





22.79 mmol m$^{-2}$ d$^{-1}$ (integrated over a 80 m ML as in Closset et al., 2014). This production is half the one measured at A3-2 from 24h incubations (46.8 mmol m$^{-2}$ d$^{-1}$ Closset et al., 2014). Such difference can be explained by the different integration time associated with the two methods the δ$^{30}$Si of exported BSi integrates one month, while a daily incubation only represents a snapshot of BSi production. The high production rate measured from 24h incubation may not have been maintained at this rate for the whole December. Additionally, DSi supply also occurs in summer (Fripiat et al., 2011; Closset et al. 2014), a process that is not included in Rayleigh equations which consequently underestimate BSi production.

(iii) Mid-summer. The sharp decrease of δ$^{30}$Si of settling particles observed in January 2012 (from 1.99 to 1.72 ‰) can be associated to a strong vertical mixing event that brings new and isotopically light DSi in the ML. Such a feature has already been observed in open ocean sediment traps time-series in the Antarctic Zone (Varela et al., 2004; Closset et al., 2015). Considering that at this time, the system behaves following an open mode (Fig. 4), such 0.27 ‰ δ$^{30}$Si$_{BSi}$ decrease would correspond to the same change in δ$^{30}$Si$_{DSi}$ that could be associated to an increase of 7.56 µmol L$^{-1}$ of the DSi concentration in the ML (1.99 and 1.72 ‰ correspond to a DSi concentration of 32.54 and 24.98 µmol L$^{-1}$, respectively). The subsequent DSi stock in the ML would be 17.06 µmol L$^{-1}$ in December 2011. Using a simplistic approach (see supplementary method) we estimate that such Si supply would require a ML deepening of ca. 37 m which is consistent with the order of magnitude of ML variation in this area (Park et al., 2008b). Indeed, high wind events could induce vertical mixings of the upper ocean entraining cold water into the ML and bringing DSi into the euphotic zone. Integrating the DSi increase of 7.56 µmol L$^{-1}$ over this new ML (80 m + 37 m) suggest a supply of 884 mmol m$^{-2}$. Such a mixing event would represent approximately 74 % of the total summer supply (1200 mmol m$^{-2}$; Closset et al., 2014) and would allow the second bloom to appear as observed in January 2012 (Rembauville et al., 2015a). The sporadic upwelling performed by Ekman pumping above the Plateau (see Gille et al., 2014) is another process that could bring DSi into the ML, but considering its low intensity, it is unlikely that it could sustain this flux of DSi. Note that if the closed model is considered, the δ$^{30}$Si$_{BSi}$ signal would correspond to a consumption of 64 % of the winter DSi stock and thus an increase of the DSi concentration of 2.3 µmol L$^{-1}$ in the ML. This value would imply a deepening of the ML by only 8.75 m and a summer supply of 204 mmol m$^{-2}$ which does not correspond to the one estimated by Closset et al. (2014). This highlights that, as proposed previously, BSi isotopic and mass balance can follow either a closed or an open mode depending on physical and biogeochemical conditions which vary during the growing season.

(iv) Late summer: The δ$^{30}$Si increased from 1.72 to 2.54 ‰ due to a second episode of enhanced biological activity in the ML. Using the Rayleigh equations (Fig. 4), this highest value corresponds to a consumption of 82 % of the winter DSi standing stock which results in a ML DSi concentration of 6.05 µmol L$^{-1}$ and thus a consumption of 10.87 µmol L$^{-1}$ of the DSi concentration in the ML between December 2011 and the end of January 2012 (52 days). Integrated over a 80 m ML, this value corresponds to a net BSi production of 16 mmol m$^{-2}$ d$^{-1}$. This production is consistent with the mean net BSi




production of 10 mmol m$^{-2}$ d$^{-1}$ estimated for the same period by Closset et al. (2014) which decreases progressively to values lower than 1 mmol m$^{-2}$ d$^{-1}$.

(v) Winter: From March to September, the $\delta^{30}$Si of settling particles remained relatively heavy but decreased gradually toward lower values as winter deep convection progressively took place above the Plateau (Blain et al., 2013).

## 4 Conclusions

The spatial distribution of $\delta^{30}$Si signatures of seawater and particulate matter in the ML was strongly impacted by the complex structure of water masses generated by the interaction between the PF and the bathymetry. In contrast, in deeper layers (> 500 m), $\delta^{30}$Si$_{DSi}$ were remarkably homogeneous (on average 1.28 ± 0.08 ‰ and 1.05 ± 0.06 ‰ for UCDW and LCDW respectively) and deep $\delta^{30}$Si$_{BSi}$ exhibited constant values in all out-plateau KEOPS-2 stations (on average 1.74 ± 0.13 ‰), suggesting that dissolution of opal did not have any significant isotopic effect during this early season.

The measured silicon properties of the ML can be considered as representative of the bloom initiation as also supported by companion studies from the same cruise (Blain et al., 2015; Cavagna et al., 2015; Closset et al., 2014; Lasbleiz et al., 2014). During this period, the Kerguelen area was characterized by a mosaic of biogeochemical environments (Trull et al., 2015). The HNLC area, strongly iron-limited (station R2), exhibited very low biomass, low BSi-production and isotopically light BSi. However, moderate production in agreement with other proxies (Closset et al., 2014; Dehairs et al., 2015; Jacquet et al., 2015) could explain the relatively high $\delta^{30}$Si$_{DSi}$ values measured at this station, precluding its application as non-productive reference station for silicon biogeochemical cycle. The iron flux in the ML above the Plateau strongly stimulated diatom production, increasing significantly its $\delta^{30}$Si$_{BSi}$ (from 0.77 ± 0.05 ‰ to 0.96 ± 0.08 ‰ over 27 days during KEOPS-2) and simultaneously enriching the ML in $^{30}$Si (up to 2.10 ± 0.05 ‰). North of the PF, stocks and Si-isotopic composition of seawater and particles were respectively lower and heavier than above the Plateau in accordance with its PFZ characteristics as already reported by Cardinal et al. (2007, 2005), Varela et al. (2004) and Fripiat et al. (2011b) in other sectors of the Southern Ocean. The situation was different in the Meander since these stations received only moderate and sporadic iron supplies. There, non-optimal light-mixing regime and nutrients availability (including iron) delayed the bloom development and lead the system to behave near steady-state with low DSi utilization. Indeed, both Si isotopic signatures and mass balance did not evolve significantly in the Meander ML.

Our Si-isotopic data allow identifying initial conditions before diatom growth, Si sources, and the connections between different water masses through local circulation. The same Si source (WW Plateau and not WW HNLC as initially thought) can be applied for both Plateau and Meander stations suggesting that water masses and the ML bloom originated from the



southeast part of the Kerguelen Plateau and spread northward. The Si-properties of this source ($31.5 \pm 2.2$ µmol Si L$^{-1}$ and $1.76 \pm 0.03$ ‰) allow us to refine the seasonal net BSi production at $3.0 \pm 0.3$ mol Si m$^{-2}$ y$^{-1}$ above the Plateau (instead of $10.5 \pm 1.3$ mol Si m$^{-2}$ y$^{-1}$ estimated by Fripiat et al., 2011a), which is more consistent with the published values for the AZ (2.4 to 3.3 mol Si m$^{-2}$ y$^{-1}$; e.g. Pondaven et al., 2000; Nelson et al., 2002). This suggests that, even if the iron-fertilization in

some regions of the Southern Ocean stimulates the uptake of C and N and the production of organic matter compared to non-fertilized area, it does not necessary enhance significantly the consumption of silicic acid and production of BSi. This observation can be explained by the decoupling between the biogeochemical cycles of Si, C and N achieved by communities of diatoms living under different biogeochemical conditions. These distinctive local communities may induce variations of the efficiency of the silicon pump (Dugdale et al., 1995) that preferentially recycles organic matter over biogenic silica in

surface waters and thus lead to moderate net BSi production despite high primary production. Moreover, above the Plateau, the WW did not have undergone any net BSi production above the plateau since their characteristics were similar over the whole season. This contrast with the meander where WW Si characteristics (concentration and isotopic composition) can vary and can be explained by successive mixings between ML and WW.

In the naturally Fe-fertilized area of Kerguelen, we have shown that the dissolved and particulate Si-pools were decoupled. In this high productive area, the strong activity, promoting vertical and lateral nutrient exchanges, drove the evolution of $\delta^{30}$Si$_{DSi}$ following steady state equations, while $\delta^{30}$Si$_{BSi}$ followed more the Rayleigh equations. Depending on the regime of BSi export that operates (e.g. BSi accumulation in the ML in early spring vs. massive export event at the end of the productive season), the ML $\delta^{30}$Si$_{BSi}$ can be alternatively described by the accumulation or instantaneous product, or could lie

between these two ideal situations. If confirmed in other productive regions of the Southern Ocean, this observation could have great implications for paleoceanographic studies. Indeed, the $\delta^{30}$Si$_{BSi}$ is currently used as a proxy for past reconstructions of surface Si-utilization (e.g. De la Rocha et al., 1998; Ehlert et al., 2013). In these studies, the isotopic signature of sedimentary opal is related with the $\delta^{30}$Si$_{DSi}$ and the extent of DSi consumption in surface waters by using either the Rayleigh's or steady state's equations without taking into account this possible decoupling between $\delta^{30}$Si$_{DSi}$ and $\delta^{30}$Si$_{BSi}$

and its seasonal variability.

Finally we have identified and quantified the processes that control the biogeochemical cycles of silicon the ML over near one complete year. We show that the $\delta^{30}$Si of settling diatoms collected in sediment traps is a powerful proxy to quantify Si fluxes in the ML, such as silicic acid consumption in spring and summer or Si supply to ML during mixing events.

**Acknowledgements.** The authors would like to thank B. Quéguiner as the KEOPS-2 chief scientist, the captain and the crew of the R/V Marion-Dufresne II for assistance on board. The research leading to these results on silicon has been funded by the European Union Seventh Framework Programme under grant agreement n°294146 (MuSiCC Marie Curie CIG).



KEOPS-2 was supported by the French Research program of INSU-CNRS LEFE-CYBER ('Les enveloppes fluides et l'environnement' – 'Cycles biogéochimiques, environnement et ressources'), the French ANR ('Agence Nationale de la Recherche', SIMI-6 program), the French CNES ('Centre National d'Etudes Spatiales') and the French Polar Institute IPEV (Institut Polaire Paul-Emile Victor).

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

**Figure captions**

Figure 1: Map of the KEOPS-2 cruise area (Indian sector of the Southern Ocean) showing the location of stations discussed in this study and the general surface circulation (black arrows from Park et al., 2014). Colours represent the detailed map of the satellite-derived surface chlorophyll a concentration (MODIS level 3 product) averaged over the cruise period. Black lines are 500 m and 1000 m isobaths.

Figure 2: Vertical profiles of biogenic silica and silicic acid concentration (a. and b.; μmol L$^{-1}$) and isotopic composition of dissolved and particulate silicon (c. and d.; ‰) for the main contrasted KEOPS-2 stations (A3, R2, E4W and F-L) and for the Meander stations (TNS6, E1, E2, E3, E4E and E5).

Figure 3: Seasonal evolution of the isotopic composition of dissolved (a.) and particulate silicon (b.) in the upper 500 m of the iron-fertilized and HNLC reference stations. Summer isotopic signatures (KEOPS-1, Fripiat et al., 2011a) are in grey

and black colours; results from the spring period (this study) are in red and blue colour.

Figure 4: Silicon isotopic composition vs. silicic acid concentration for the different ML Si-reservoir in the iron-fertilized area of the Kerguelen Plateau. A unique source (TNS6-WW, blue dot) was used for all Plateau and Meander stations. ML dissolved Si-pools above the Plateau (purple dots) and in the Meander (red dots), fit well with a steady state fractionation law (grey straighted line) but not with a Rayleigh distillation law (grey broken line). Grey dotted line represent the 0.2 ‰ sd

of the fractionation factor. ML particulate Si-pools above the Plateau (black dots) and in the Meander (green dots) lie between the instantaneous and the accumulated products predicted by a Rayleigh distillation model (dotted and broken lines respectively). Straight black line represent the products in the steady state model. Summer isotopic signatures (K1 for A3 KEOPS-1) are from Fripiat et al. (2011a).

Figure 5: Potential mixing curves between TNS6-WW and the different Meander ML water masses, WW are in blue and

ML are in red. The two end-members were identified by black curves. The two grey dashed lines represent the mixing curves between TNS6-WW and TNS6-ML and TNS6-WW and E1-ML which may explain the Si properties of E1-WW and E2-WW, respectively . The ML averages and error bars were calculated based on the median values and interquartile range for each station while they were the « central value » defined using the salinity threshold method for WW.

Figure 6: Surface chlorophyll a concentration measured by satellite (green line), BSi fluxes collected in the sediment trap

at 289 m (blue bars), and silicon isotopic composition of settling diatoms (black diamonds) at the A3 station.





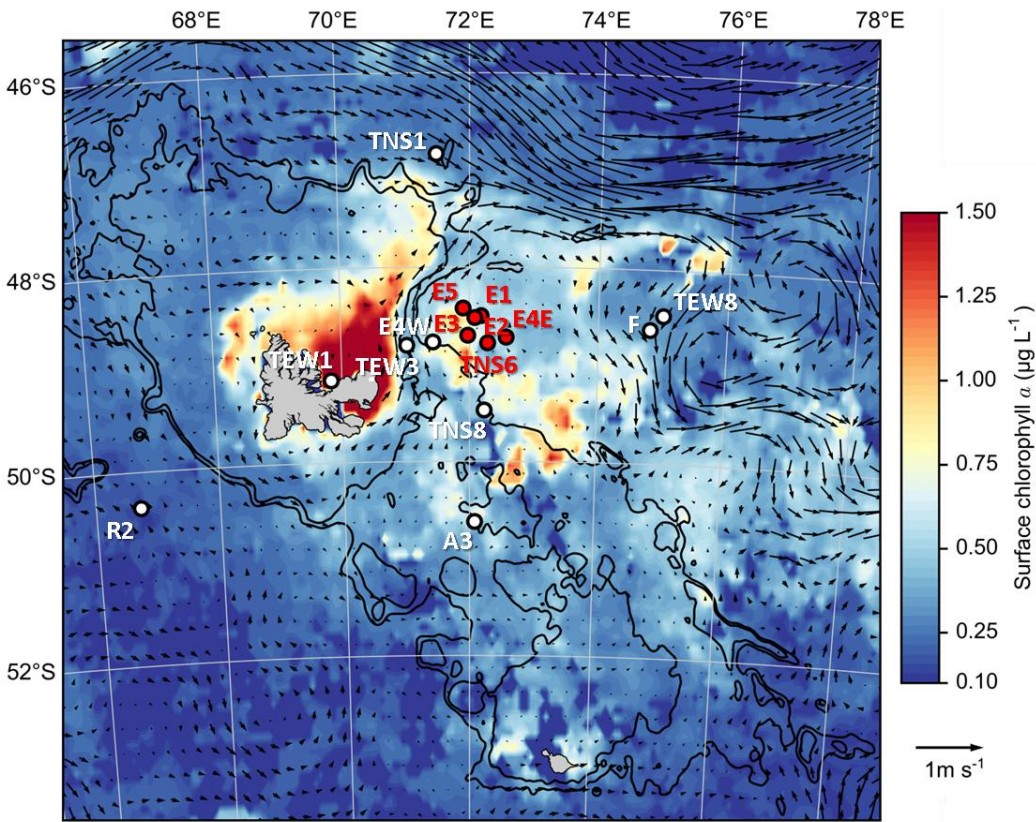

**Figure 1**





**Figure 2**





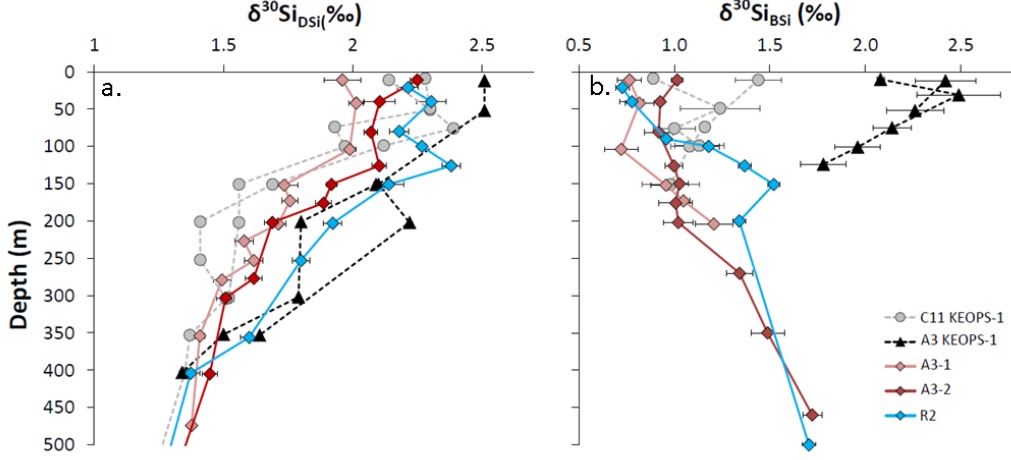

**Figure 3**

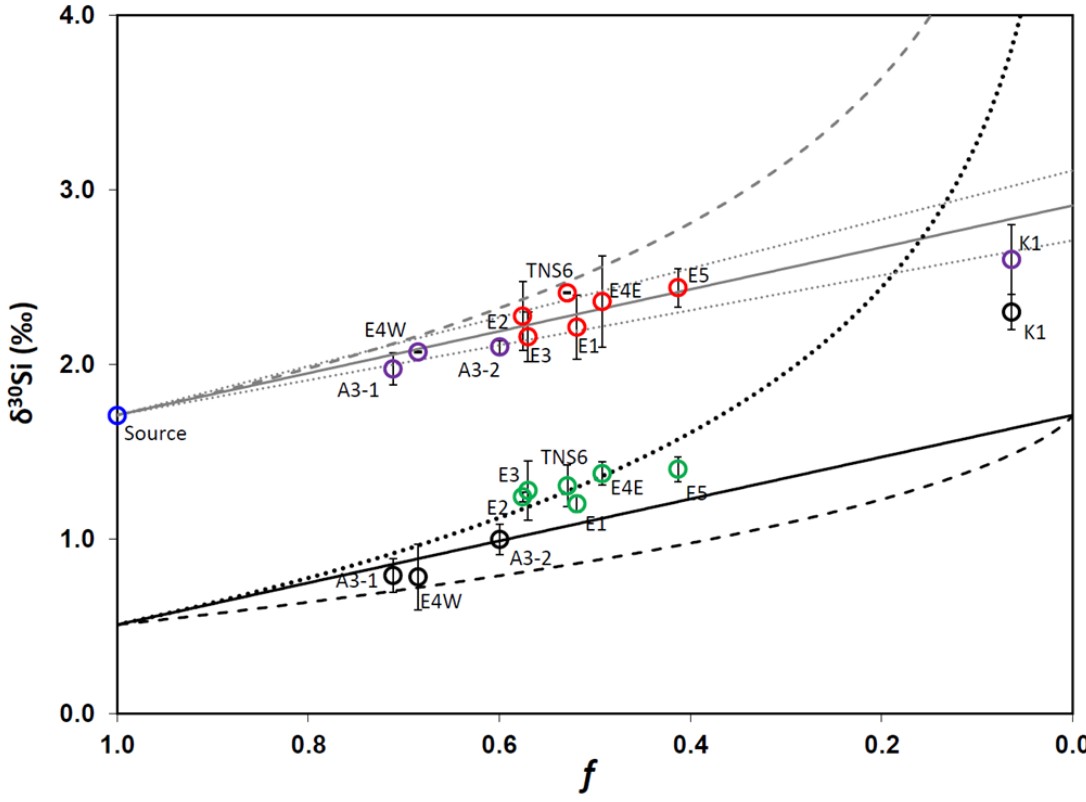

**Figure 4**





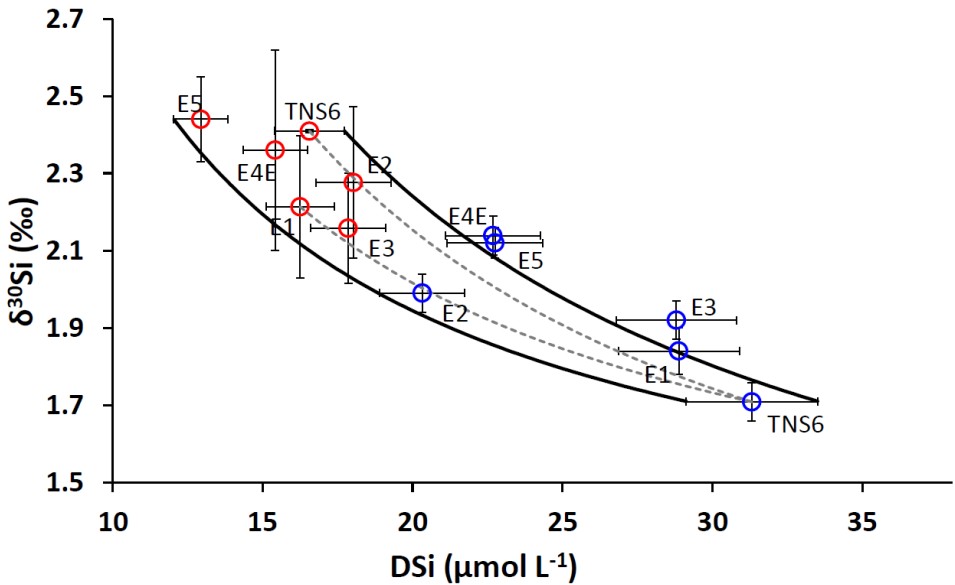

**Figure 5**

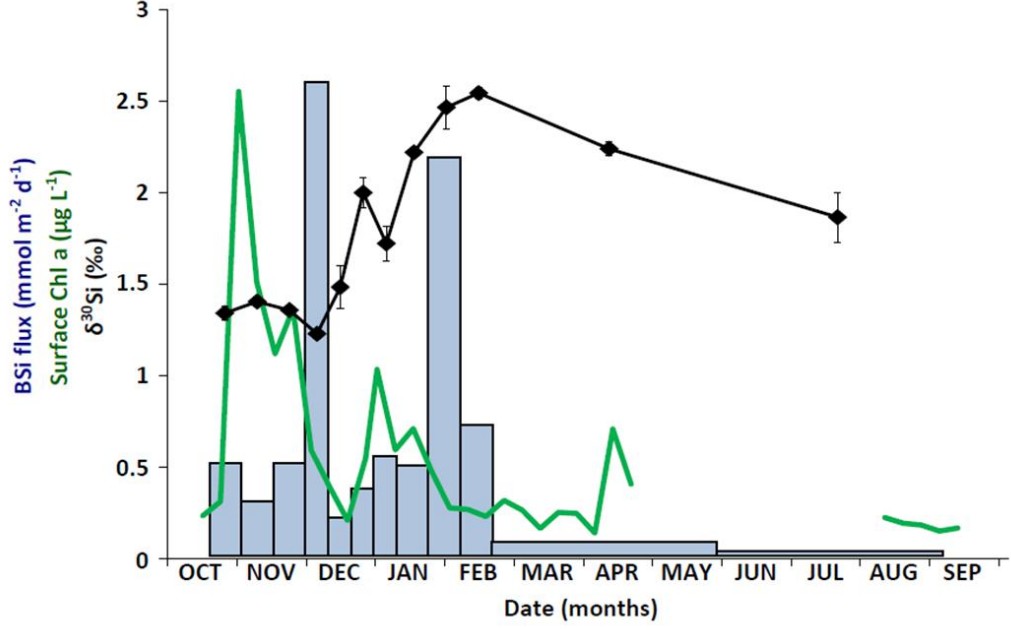

**Figure 6**