# Peer review of "Unveiling the Si cycle using isotopes in an iron fertilized zone of the Southern Ocean: from mixed layer supply to export"

_Biogeosciences, 2016_

## Referee Comment (RC1) · Anonymous Referee #1 · 27 Jun 2016

See attached text

Please also note the supplement to this comment:
http://www.biogeosciences-discuss.net/bg-2016-236/bg-2016-236-RC1-supplement.pdf

————————————————————

---

## Referee Comment (RC2) · Anonymous Referee #2 · 19 Jul 2016

**Review of manuscript BG-2016-236** *Gregory de Souza*

In their manuscript, Dr. Closset and colleagues present an impressively large dataset of silicon stable isotope data ($\delta^{30}$Si) of both dissolved and particulate silica from the region of natural iron fertilization around the Southern Ocean island of Kerguelen. Analytical and sample processing methods are very well documented and data quality is discussed in detail, providing confidence that the presented data are of high quality.

Using their data, the authors examine the isotope dynamics of Si production and export in this oceanographically complex and biogeochemically interesting region. They show systematic biogeochemical and isotopic evolution in the iron-fertilised region examined and improve previous $\delta^{30}$Si-based estimates of Si supply to and export from the surface mixed layer. Comparison of collocated dissolved and biogenic silica allows the authors to study the Si isotope dynamics of the region in some detail, and they discuss the complex behaviour they observe. Finally, analysis of samples collected by a year-long deployment of a sediment trap provides insights into the seasonal cycle of export in this region and how this is reflected in the isotopic composition of sinking opal.

All in all, I find this holistic isotopic view of Si cycling around Kerguelen interesting and certainly worthy of publication in *Biogeosciences*. The study is well conceived, and its aims are clearly laid out and tackled. However, I found the Discussion section rather hard to read, and I think that the manuscript would profit from an effort to make the profusion of stations and data more easily understandable for the reader; without this, it is hard to understand the manuscript's main points.

Scientifically, I have three main issues with the manuscript: (a) there are some conceptual inconsistencies in the interpretation of mixed-layer isotope systematics (Section 3.3.3) that must be corrected, (b) there are errors in plotting depth profiles of $\delta^{30}$Si$_{BSi}$ (Fig. 3h) that may have affected the authors' interpretation of the data, and (c) the equations used to calculate quantitative results in Section 3.3.1 and 3.3.4 are not given. I detail these points below, together with more minor points that the authors should also consider while carefully revising this manuscript.

*General comment: Manuscript organization*

Currently, the manuscript requires very careful reading (and re-reading) in order to understand the authors' argumentation and get a sense of the various settings. I think the authors would do well to try to make the data more accessible to readers unfamiliar with the details of KEOPS-2. A few suggestions:

- I would spend some time at the beginning of the Discussion (rather than in the Methods section) to introduce the groups of stations and their oceanographic settings, and make sure to refer to these settings consistently (currently, the text sometimes refers just to station numbers, sometimes to stations "north of the PF" even though this has not been referred to before, etc.). Such clear and consistent nomenclature is vital for the reader to be able to follow the authors' reasoning easily.
- The groupings used in the text could be shown schematically on Fig. 1, i.e. show the "HNLC" region, the "Plateau" region and the "Meander" region in different shadings. Also, Fig. 1 would benefit from a more schematic representation of the flow, especially together with frontal positions.
- The timing of sampling is key to the authors' seasonal interpretation of the data but is not referred to very clearly: e.g. given their similarities in WW Si properties and use as the seasonal Si source, it is very important to know that TNS06 and A3-1 were sampled just 2 days apart. But this information is only visible in the supplementary information.

**Major comment 1: Interpretation of isotope systematics**

In Section 3.3.3, the authors discuss the mixed layer isotope systematics of the plateau and meander regions on the basis of the Rayleigh and steady-state models. Clearly, the isotopic behaviour in this region is complex, and the authors' discussion does justice to this, taking into consideration the various processes that may be affecting the isotope dynamics. However, I have a fundamental problem with the interpretation that BSi follows Rayleigh systematics whilst DSi does not – this simply cannot work. The evolution of the Rayleigh product (instantaneous or accumulated) is intimately tied to the evolution of

the DSi pool – i.e. the product cannot follow the Rayleigh curve unless the DSi pool *also* experiences the strong fractionation associated with closed-system systematics. Thus whilst there is clearly complexity in the isotopic dynamics here, the authors interpretation of "decoupled" systematics (Rayleigh for BSi, steady-state for DSi) cannot be correct. Could the results instead be due to a difference in the timescales over which the two sampled pools are integrating? Regardless of the specific reason, the author's analysis shows that whilst the model framework is useful for interpreting isotopic systematics, it also has limitations when applied to the real ocean, and I would recommend a more careful interpretation that is less strictly dependent on conformance to idealised model curves.

Also, there is an inconsistency between the text and the caption of Fig. 4 regarding the Si source considered for the analysis. The text (L28 on p12) mentions "averaged Plateau-WW", whilst the figure caption mentions "TNS6-WW".

**Major comment 2: Error in plotting $\delta^{30}Si_{BSi}$ data**

Based on the data presented in the supplementary material, the depth axis of Fig. 3h appears to be wrong. This does not allow a direct comparison of BSi concentration and isotope composition for the meanders stations and makes it hard to assess the author's interpretation of these data in Section 3.2.3, since it is not clear whether their interpretation is based on the faulty figure or not. This must be corrected.

Regardless of this error, the authors' argument for accumulation in the deeper ML or just below does not seem to be borne out by the data, given that BSi concentrations decrease strongly below 100m. Given the many references to WML and ML in the discussion here and elsewhere, I think it would be useful to show physical parameters (or at least MLD) for the stations. Barring that, it is difficult for the reader to follow the authors' reasoning.

**Major comment 3: Calculation of quantitative results**

In Sections 3.3.1 and 3.3.4 respectively, the authors calculate seasonal net BSi production and estimate surface DSi concentrations from isotope data, without giving details of the equations used (Section 3.3.1) or the assumptions made (Section 3.3.4) in order to achieve the results. Whilst I appreciate that the authors have done this before in other publications, it would be much better if their methods used for this study were documented here as well.

**Minor comments**

- *Meander WW evolution:* In Section 3.3.2, the authors discuss the evolution of WW $\delta^{30}Si$ in the meander region. Whilst it is possible that diapycnal interaction is responsible for this evolution as the authors argue, clearly the possibility of lateral interactions across the PF must be considered as well. After all, the intense mesoscale activity at the front acts strongly to stir/exchange tracers *laterally/isopycnally* as well as mixing them diapycnally (e.g. Dufour et al. 2015, *J. Phys. Oc.*, 10.1175/JPO-D-14-0240.1) . As the authors argue for the ML in Section 3.2.3, lateral mixing of Si-poor PFZ subsurface waters with WW could thus well produce the evolution observed. In Fig. 5, it would be interesting to compare the evolution expected from such lateral subsurface mixing with that expected from diapycnal mixing.
- *Strange HNLC station:* In my opinion, not enough reference is made to the fact that the HNLC station has a unique isotopic behaviour – its $\delta^{30}Si_{DSi}$ evolves to very heavy values whilst $\delta^{30}Si_{BSi}$ is very light, suggesting a very different expression of fractionation here. The authors explanation of the $\delta^{30}Si_{BSi}$ data from this station (L25-21, p9) is unclear and thus not convincing.
- *Silicate pump:* In Section 4 (L7-10, p19), I do not understand how the silicate pump could lead to low BSi production despite high primary productivity. Should it not be the other way around if silica is more efficiently exported than organic matter? As a mechanism of decoupling Si, N and C, I am surprised that no mention is made of the Fe-related plasticity in Si:N ratios (as shown by the canonical 1998 papers by Sunda and Huntsman or Takeda), which I think would explain direction of change better.
- *Various minor comments:*

- *UCDW/LCDW identification* (L3-5, p5): It is UCDW that is associated with the $O_2$ minimum, not LCDW (e.g. Talley 2013, *Oceanography* 26, 80-97)
- *Sections 2.2 and 2.3* both have the same heading. These should be "Sample collection" and "Sample preparation" respectively. The sub-sections of Section 2.3 are numbered incorrectly. On L6, p6, could there be an error in the units given for NaOH concentration?
- *Section 2.5*: it would be better to refer to "full external" rather than "global" reproducibility.
- *Section 3.1*: Cao et al. 2012 is not a Southern Ocean study.
- *Section 3.3 and Fig. 3*: The comparison with late summer conditions during KEOPS-1 is useful, but Fig. 3 is confusing since it combines the seasonal signal (i.e. KEOPS-1 vs. KEOPS-2) with a regional/biogeochemical signal (i.e. HNLC versus Plateau). The figure should separate these regions into separate panels and also show concentration profiles to provide context for the isotope data.
- *Section 3.3.4*: It would be good to mention once again that the sediment trap is located at Station A3. On L33, p16, I don't understand why the ML concentration estimated from sediment trap data is compared with mid-November surface concentrations at station E5, rather than with mid-November concentrations at the station where the sediment trap is located (A3-2). Concentrations here are about 2x higher than estimated from the sediment trap $\delta^{30}Si_{BSi}$ data. Also, on L29, p17, it should be mentioned that the high values seen in late summer cannot be explained by the steady-state model.
- Finally, although in general the manuscript's English is good, it would profit from being proof-read by a native English speaker in order to iron out small grammatical errors.

---

## Author Comment (AC1) · 15 Aug 2016

Please find enclosed the answers of all referee's comments

Please also note the supplement to this comment:
http://www.biogeosciences-discuss.net/bg-2016-236/bg-2016-236-AC1-supplement.pdf

---

## Author Response (AR2)

Dear editor,

We would like to thank you for your advices about the general organisation of the manuscript, and to thank both referees for their constructive and helpful comments on our manuscript. We detail below point by point how we plan to revise the article for publication in Biogeosciences.

**Anonymous Referee #1**

My main concern is about the utilization of the steady state model ('Open') to describe a system where there is a seasonal nearly-complete DSi consumption (Kerguelen Plateau). This model assumes steady state conditions in the mixed layer and, therefore, no changes in concentrations with time.

It is correct that the steady state model when it reproduces a flow-through reactor assumes a continuous supply of substrate and thus no change in concentration with time. Theoretically, a series of sequential steady state models would reproduce finally the isotope dynamics of a closed system and allow consumption of substrate (Fry, 2006). It appears that in some oceanic Si or N isotopic systems, the steady state reproduces better the data (e.g. Sigman et al. 1999; Cardinal et al. 2005) than closed Rayleigh model. More particularly, in the case of the Kerguelen Plateau, Fripiat et al. (2011) have already shown that the steady state model better describes the seasonality of the silicon isotopes in the ML and it seems that our data confirm this observation. In their study, authors have suggested that there was a significant ventilation of the ML above the Plateau which supply DSi with low  $\delta^{30}$ Si value into the system. We acknowledge that it is conceptually hard to reconcile a seasonal decrease of concentration and increase of isotopic composition with the steady state but so far, no better model has been shown. In the revised version, we will make clearer this apparent contradiction and raises that we can consider each steady-state observed in the area as a snapshot highlighting a Si supply from below and that the steady state must be broken and formed again when surface concentration is decreasing (i.e. when supply does not exactly compensate uptake)

Authors also look at figure 4 to infer which models fit the most with the observations. I'm not sure that the level of precision is sufficient to distinguish between models. Especially if we take into account the precision on both the fractionation factor and the Si-source: on figure 4, the former is shown only for the steady state model and the latter is not taking into account.

We didn't add the precision on the fractionation factor for the Rayleigh model to avoid overloading the figure but we will add it on fig.4 as for the steady state model. When adding this uncertainty, the productive stations above the plateau (A3-2, E5 and K1) remain clearly out of the Rayleigh fractionation trend. The isotopic composition and the DSi concentration of the Si-source are much better constrained  $(\pm 2.2 \mu mol L^{-1} and \pm 0.03 \%$ , respectively) in this study and do not change the fact that Rayleigh model – contrary to steady state – cannot explain the measured isotopic values especially at high consumption in the late summer. We will mention this in the text of the revised version.

In late summer, the authors also infer that the biogenic silica pool is a mixture between new (high- $\delta$ 30Si) and old (low- $\delta$ 30Si) biogenic silica (from figure 4). But late summer DSi  $\delta$ 30Si is low (figure 4), and actually not significantly different than biogenic silica  $\delta$ 30Si (error bars in figure 4). How is it possible to produce high- $\delta$ 30Si biogenic silica from a low- $\delta$ 30Si pool? I agree that biogenic silica  $\delta$ 30Si could be explained by being a mixture between instantaneous and accumulated products in the Rayleigh model ('closed'). But in regard of the figure 4, assuming a constant fractionation factor, such high- $\delta$ 30Si biogenic silica requires a DSi pool being significantly higher than described with the steady state model at any time, or than the observations.

The isotopic dynamics in this region is complex regarding the various processes that can affect them. Several explanations are possible to describe what we observed, but it is difficult to discriminate which one is the more relevant:

- The fractionation factor of diatoms ( ${}^{30}\varepsilon$ ) may vary over the season. Since there is no study yet that focuses on  ${}^{30}\varepsilon$  seasonal variations (only one study has highlighted interspecific – but not seasonal – variations, see Sutton et al., 2013), we have considered it as constant, but we are aware that this may not reflect the reality in the ocean and that the constancy of the fractionation factor can be challenged.

- In the ocean, the distinction between steady state and Rayleigh models can be tricky. Indeed, many situations, such as the Kerguelen region, can combine or switch between characteristics of both closed and open system dynamics resulting in changes of the apparent fractionation factor ( $\Delta^{30}Si = \delta^{30}Si_{DSi} - \delta^{30}Si_{BSi}$ ). As already proposed by Fripiat et al. (2012) we suggest that these variations are mainly controlled by the Si uptake:Si supply ratio of the system. When Si uptake:Si supply ratio is high (during the bloom period), the system roughly follows a Rayleigh fractionation model. However, when the DSi pool limits Si uptake (at the end of the productive period), the Si uptake:Si supply ratio decreases and the supply of light DSi into the ML by vertical mixing decreases significantly the  $\delta^{30}Si_{DSi}$ . The  $\delta^{30}Si_{BSi}$  is affected differently since the BSi pool is composed by a mixing between newly formed (and light) diatoms and isotopically heavy diatoms that have been produced previously in the ML. Such process would explain why we observe late summer high  $\delta^{30}Si_{BSi}$  associated to low  $\delta^{30}Si$  of the DSi pool.

- Other processes such as mixing or BSi dissolution with or without isotopic fractionation, can affect differently the different pools and would lead to variations of the apparent fractionation factor. Mixing brings light DSi into the ML and decreases instantaneously the  $\delta^{30}$ Si of the DSi pool while the dissolution of biogenic silica may affect mainly light and dead diatoms that were produced in the beginning of the bloom resulting in a progressive increase of the  $\delta^{30}$ Si of the BSi pool. Such effects are temporally decoupled and would result in a decoupling of the information recorded in the dissolved and particulate phase (see also our reply to main concern of reviewer 2).

Hence even if these theoretical models seem to be appropriate for interpreting paleoceanographic isotope records that integrate longer temporal scale, it appears that they have limitations when applied to the modern ocean at shorter and seasonal scales.

We agree that this part of the discussion if confusing. We will rearrange the text to make it clearer.

**Minor comments:**

Pages 8-9 Lines 32-1: What is real WW Si properties? As written, it is not clear.

In this sentence, we will change "real WW Si-properties" by "specific Si concentration and isotopic composition of the WW".

Page 10 Lines 28-29: Not sure to understand why the location of the meander being upstream of the Kerguelen Plateau is indicative of a delay in the initiation of the bloom.

As shown in Fig.1, and explained in the text, the meander is located downstream of the Plateau. Water comes from the South (see Park et al., 2014 and Sanial et al., 2015) and undergoes a progressive phytoplankton bloom and consumption of nutrients.

Page 11 Lines 9-11: Such accumulation is not seen on BSi concentration profiles? In addition, an accumulation should imply low biogenic silica  $\delta$ 30Si, no?

Yes, we realize that the interpretation here is inadequate. Actually we do not see subsurface accumulation of heavy BSi here, but a production of light BSi in surface. This lower surface BSi isotope composition could result from the supply of light DSi into the ML by sporadic mixing events. The DSi stock in the ML would have a lower isotopic signature than previously and diatoms would produce lighter BSi at the top of the ML. We will change our discussion (from line 6 to line 18) according to this.

Page 11 Line 30: The authors need to define better the concept behind "new silicic acid" and its relationship with silicon isotopes. As written, it is not clear.

**We will remove "new" in this sentence as it can generate confusions with other parts of the discussion.**

Page 12 Lines 1-2: the northern Kerguelen shelf...(to be added)...or from the mixing with waters north of the polar front bearing higher DSi  $\delta$ 30Si due to the progressive export of low BSi  $\delta$ 30Si along the meridional overturning circulation path. From my point of view (given the circumpolar boundary between APF and PFZ), this should be the dominant driver of such increase in subsurface DSi  $\delta$ 30Si.

We agree, this is what we wanted to say. We will replace our sentence by the referee's one and add "including from coastal waters".

Page 12 Line 18: HNLC area in KEOPS 1&2 are located in different areas, East and West of the Kerguelen Plateau. Are they really comparable? It should be discussed somewhere when you highlight the differences and make some hypotheses on the water sources. For example, HNLC WW during KEOPS 1 appear to be actually more representative (from their geographical locations) of this expected source of waters originating from the South. But I agree that measurements show the opposite, as being more similar to the HNLC area west of the Kerguelen Plateau. So, I do not really understand how the authors (latter in this paragraph) infer a source for the Kerguelen Plateau WW coming from the South based on these observations.

We agree that HNLC area in KEOPS 1 and 2 cannot be compared and be used as water sources for the model. We discuss that later (line 23-24 p12). Actually we did not choose a source coming from the South based on the observations. We have defined our source as the WW above the Plateau and discuss that this water mass likely originates from the South (based on hydrological and other biogeochemical observations, see Park et al., 2014 and Sanial et al., 2015). The Si isotopic composition and concentration of this water mass source in the HNLC WW might have been modified during its transport

in subsurface from HNLC to the plateau (e.g. by sporadic mixing as seen in the region) and before being supplied to the ML of the Plateau. This could explain why the WW HNLC seems to be inadequate to characterize Si concentration and isotopic composition of the Si supplied in the ML over the Plateau. In the revised version, we'll more clearly explain this.

Figure 3: DSi concentration between KEOPS1 & 2 would be useful here.

We will add DSi concentrations in this figure as suggested.

Page 13 Line 4: In the next paragraph, the authors indicate that Closset et al. (2014) did not report measurements in the deep silica maximum. There are other studies reporting direct measurements of high D/P in deep silica maximum of the Southern Ocean.

Our wording was not clear in the previous version. Actually Closset et al. (2014) did measure BSi production but they are close to 0 in the deep silica maximum, and thus high D/P ratios. High D/P ratios at depth have also been measured by Fripiat et al. (2011) in the PFZ of the Australian sector of the Southern Ocean. We will clarify this and add this reference to the text.

Page 14 Line 11: But not for biogenic silica  $\delta$ 30Si, following more closely the Rayleigh trends.

We are sorry, but we did not understand this comment. All the delta values discussed in this section 3.3.2 on WW correspond to DSi and not BSi. The  $\delta^{30}$ Si of BSi that follow more closely a Rayleigh trend, corresponds to the ML pool (and not BSi in the WW).

Figure 4: Why the error propagation for the Rayleigh model is not shown? If presented, I expect that most of the DSi  $\delta$ 30Si would still fit with the Rayleigh model. Especially if you take also into account the uncertainties related to the initial Si-source.

As previously discussed, we have excluded the precision on the fractionation factor for the Rayleigh model to avoid overloading the figure. When adding this uncertainty, the productive stations above the plateau (A3-2, E5 and K1) remain clearly out of the Rayleigh fractionation trend.

Page 14 Lines 7-22: I like this paragraph. Maybe it would be worth to say that such process is analog of what is observed for DSi  $\delta$ 30Si across the ACC meridional overturning circulation.

We agree and we will mention this point in this paragraph.

Page 14 Lines 32-34: The BSi accumulation/export ratio appears to be the main driver. This should be said more strongly.

Yes, we will underline more this point.

Page 15 line 16: The authors suggest that late summer newly-formed biogenic silica presents high  $\delta$ 30Si, but the residual DSi  $\delta$ 30Si is low (Figure 4). As being the Si-source, how can you produce high- $\delta$ 30Si biogenic silica from this DSi pool (assuming constant fractionation factor)?

Please see the above answer (first page of this document). We will change the text according to this explanation and provide a clearer discussion.

Page 15 Lines 27-29: Why low Si-uptake/supply ratios and a combination of open and closed system (also implying variable Si-uptake/supply ratios) are different approaches? As defined, the open model assumes that Si-supply equals the sum of BSi accumulation/export and residual DSi (implying also low or close to unity Si-uptake/supply ratio, depending of the relative Si-utilization).

Yes, variations of Si uptake:Si ratio and the combination of open and closed model are not different approaches but are closely linked together. We will correct this part of the text.

Page 15 Lines 29-31: Why the export of biogenic silica will decrease  $\Delta$ 30Si? It should be the opposite (as discussed previously). As written it is not clear.

We have realized that this part of the discussion is not clear (as raised by the reviewer). We will remove line 27 to 31 and replace it by "Our results point out that an alternative approach to change the  $\Delta^{30}Si$ could be obtained when the BSi pool in the system switches alternatively between the Rayleigh instantaneous or accumulated product depending on the Si production:Si export ratio.".

Page 15: Lines 31-35: Same comment than above. The authors suggest that late summer newly-formed biogenic silica presents high  $\delta$ 30Si, but the residual DSi  $\delta$ 30Si is low (Figure 4).

Please see the above answer (first page of this document). We will change the text according to this explanation and provide a clearer discussion.

Page 16 Lines 19-20: Dissolution occurs in subsurface (Nelson et al., 2002, DSR). This process can therefore also explain the observed trends. The authors should be more caution before completely ruling out the dissolution isotope effect.

We agree, dissolution can occur in subsurface. But during KEOPS-2, dissolution rates measured from 0 to 80 m were very low and since we were at the beginning of the productive period, we can expect that they remain low in deep waters. Moreover, even if we can expect some dissolution of biogenic silica in the water column, some recent studies have shown that isotopic fractionation during dissolution did not affect the BSi  $\delta^{30}$ Si along the water column (Fripiat et al., 2012; Closset et al., 2015) and may not occur (Wetzel et al., 2014). This is why we have chosen to rule out the dissolution isotope effect.

Page 18 Lines 27-28: Why here the steady state model assumes, as it should be, no variations in concentrations with time, but it has been previously used to describe a situation over the Kerguelen Plateau exhibiting large variations in DSi concentration?

The steady state model has been previously used to describe the system over the Kerguelen Plateau by Fripiat et al. (2011). In their study, authors have suggested that there was a significant ventilation of the ML above the Plateau which supply DSi with low  $\delta^{30}$ Si value into the system even if they have observed a consumption of DSi in surface waters. In our study, the steady state model assumes no variations in concentrations with time and this is exactly what we have observed in spring in the Meander. Vertical and/or horizontal mixings supplied DSi in the ML that allow the system to behave following a steady state model.

**Reviewer #2**

**General comment: Manuscript organization**

Currently, the manuscript requires very careful reading (and re-reading) in order to understand the authors' argumentation and get a sense of the various settings. I think the authors would do well to try to make the data more accessible to readers unfamiliar with the details of KEOPS-2. A few suggestions:

- I would spend some time at the beginning of the Discussion (rather than in the Methods section) to introduce the groups of stations and their oceanographic settings, and make sure to refer to these settings consistently (currently, the text sometimes refers just to station numbers, sometimes to stations "north of the PF" even though this has not been referred to before, etc.). Such clear and consistent nomenclature is vital for the reader to be able to follow the authors' reasoning easily.

As requested by the editor, we will keep the description of stations in the Methods section but we will add a table to introduce the stations and describe briefly their oceanographic settings. We will also make their designation more consistently in the text.

- The groupings used in the text could be shown schematically on Fig. 1, i.e. show the "HNLC" region, the "Plateau" region and the "Meander" region in different shadings. Also, Fig. 1 would benefit from a more schematic representation of the flow, especially together with frontal positions.

*Ok, we will simplify the figure and make the distinction between "HNLC", "Plateau", "Meander" and "North of the Front" stations in Fig. 1. Note that in order to be consistent with the many other articles from the same KEOPS-2 cruise (including a special volume in Biogeosciences) we need to keep the same labels for the stations.*

- The timing of sampling is key to the authors' seasonal interpretation of the data but is not referred to very clearly: e.g. given their similarities in WW Si properties and use as the seasonal Si source, it is very important to know that TNS06 and A3-1 were sampled just 2 days apart. But this information is only visible in the supplementary information.

We will include this information in the table that will describe the stations (please see previous comment).

**Major comment 1: Interpretation of isotope systematics**

In Section 3.3.3, the authors discuss the mixed layer isotope systematics of the plateau and meander regions on the basis of the Rayleigh and steady-state models. Clearly, the isotopic behaviour in this region is complex, and the authors' discussion does justice to this, taking into consideration the various processes that may be affecting the isotope dynamics. However, I have a fundamental problem with the interpretation that BSi follows Rayleigh systematics whilst DSi does not – this simply cannot work. The evolution of the Rayleigh product (instantaneous or accumulated) is intimately tied to the evolution of the DSi pool – i.e. the product cannot follow the Rayleigh curve unless the DSi pool also experiences the strong fractionation associated with closed-system systematics. Thus whilst there is clearly complexity in the isotopic dynamics here, the authors interpretation of "decoupled" systematics (Rayleigh for BSi, steady-state for DSi) cannot be correct. Could the results instead be due

to a difference in the timescales over which the two sampled pools are integrating? Regardless of the specific reason, the author's analysis shows that whilst the model framework is useful for interpreting isotopic systematics, it also has limitations when applied to the real ocean, and I would recommend a more careful interpretation that is less strictly dependent on conformance to idealised model curves.

We totally agree and we understand it was unclear from our previous version. Indeed the most likely explanation for DSi pool to follow steady-state and BSi to follow Rayleigh is the different timescales they integrate as suggested by the reviewer. This as already raised previously by other authors (e.g. Cardinal et al. 2007; Fripiat et al., 2011). As already suggested by reviewer #1, we will rearrange the text to make it less confusing for the reader. Please see previous comment on the fractionation factor (1st page of this document) where we give a potential explanation of this apparent decoupling between the particulate and dissolved pool.

Also, there is an inconsistency between the text and the caption of Fig. 4 regarding the Si source considered for the analysis. The text (L28 on p12) mentions "averaged Plateau-WW", whilst the figure caption mentions "TNS6-WW".

We will correct the "TNS6-WW" by "averaged Plateau-WW".

**Major comment 2: Error in plotting δ30SiBSi data**

Based on the data presented in the supplementary material, the depth axis of Fig. 3h appears to be wrong. This does not allow a direct comparison of BSi concentration and isotope composition for the meanders stations and makes it hard to assess the author's interpretation of these data in Section 3.2.3, since it is not clear whether their interpretation is based on the faulty figure or not. This must be corrected. Regardless of this error, the authors' argument for accumulation in the deeper ML or just below does not seem to be borne out by the data, given that BSi concentrations decrease strongly below 100m. Given the many references to WML and ML in the discussion here and elsewhere, I think it would be useful to show physical parameters (or at least MLD) for the stations. Barring that, it is difficult for the reader to follow the authors' reasoning.

We thank reviewer 2 for his careful look at our data. We apologise for this error which was on the depth axis on Fig. 3h. As suggested by reviewer #1, we will also change the discussion regarding the BSi accumulation in this part of the text and we will add the MLD in each panels of Fig. 3.

**Major comment 3: Calculation of quantitative results**

In Sections 3.3.1 and 3.3.4 respectively, the authors calculate seasonal net BSi production and estimate surface DSi concentrations from isotope data, without giving details of the equations used (Section 3.3.1) or the assumptions made (Section 3.3.4) in order to achieve the results. Whilst I appreciate that the authors have done this before in other publications, it would be much better if their methods used for this study were documented here as well.

In Section 3.3.1 we used the same equations as those used in Fripiat et al., (2011). We will add the equations in the text or in the supp. material.

In Section 3.3.4, as written p16 L31 or p17 L31, we calculate the net BSi production in early and late summer assuming that the system is described by Rayleigh equations. The assumptions behind that are the following:

- The fractionation factor  ${}^{30}\varepsilon$  is constant.

- There is no significant BSi export during this period so the BSi export:BSi production ratio is very low.

- There is no significant BSi supply during this period so the Si supply:Si uptake ratio is very low. For example, in early summer, these two last assumptions are very consistent with the situation when the bloom has just started, when diatoms accumulates in the ML and has not consumed all the DSi

stock.

In Mid-summer, the sharp decrease observed in our data can be associated to a mixing event. As specify p17 L11, we assume that the system is describe by steady state equations. The assumption behind that are the following:

- The fractionation factor  ${}^{30}\varepsilon$  is constant.

- The DSi supplied into the system has the same isotopic signature as the source. The DSi source for the Plateau stations is the WW. Since the mixing event bring Si into the ML from deeper water, this assumption is clearly realistic.

- All the BSi and the remaining DSi should be removed from the system. We cannot verify these assumptions but we can expect that a large part of the BSi was exported out of the ML Mid-summer. This assumption might be supported by the deep silica maximum observed late summer below the ML (150 m) by Mosseri et al. (2008) at the same station.

We will list these assumptions in the revised text.

**Minor comments**

- Meander WW evolution: In Section 3.3.2, the authors discuss the evolution of WW  $\delta$ 30Si in the meander region. Whilst it is possible that diapycnal interaction is responsible for this evolution as the authors argue, clearly the possibility of lateral interactions across the PF must be considered as well. After all, the intense mesoscale activity at the front acts strongly to stir/exchange tracers laterally/isopycnally as well as mixing them diapycnally (e.g. Dufour et al. 2015, J. Phys. Oc., 10.1175/JPO-D-14-0240.1). As the authors argue for the ML in Section 3.2.3, lateral mixing of Sipoor PFZ subsurface waters with WW could thus well produce the evolution observed. In Fig. 5, it would be interesting to compare the evolution expected from such lateral subsurface mixing with that expected from diapycnal mixing.

The diapycnal mixing is sufficient to fully explain the Si characteristics of WW at stations E1 and E2 but cannot explain what we observed at stations E3, E4E and E5. We agree that a combination of vertical mixing and lateral advection is likely the reason of these observations and we will discuss this argument at the end of this paragraph with the reference to Dufour et al. (2015). However we will probably not change Fig. 5 as, considering our dataset, it is much more difficult to find the appropriate end-members for diapycnal advection than for vertical mixing and we think that we may not have enough measurements to do that properly (we would need more stations North of the polar front).

- Strange HNLC station: In my opinion, not enough reference is made to the fact that the HNLC station has a unique isotopic behaviour – its  $\delta$ 30SiDSi evolves to very heavy values whilst  $\delta$ 30SiBSi is very light, suggesting a very different expression of fractionation here. The authors explanation of the  $\delta$ 30SiBSi data from this station (L25-21, p9) is unclear and thus not convincing.

We are sorry, we realized that there was an error in fig.2b. The blue profile that evolves toward very heavy  $\delta^{30}$ Si values corresponds to station F and not R. This isotopic behavior is well consistent with the low DSi concentrations measured at station F. We will correct that.

- Silicate pump: In Section 4 (L7-10, p19), I do not understand how the silicate pump could lead to low BSi production despite high primary productivity. Should it not be the other way around if silica is more efficiently exported than organic matter? As a mechanism of decoupling Si, N and C, I am surprised that no mention is made of the Fe-related plasticity in Si:N ratios (as shown by the canonical 1998 papers by Sunda and Huntsman or Takeda), which I think would explain direction of change better.

As discussed in the text, the different diatom communities living under different biogeochemical conditions lead to variations in the silicon pump efficiency. These communities may have different degrees of silicification, explaining how we can observe high primary production but only moderate BSi production. By measuring Si uptake rates during the KEOPS-2 cruise, Closset et al. (2014) have suggested that these differences may not be primarily controlled by iron limitation such as already shown in artificial iron enrichment experiments (Takeda, 1998 or Hutchins and Bruland, 1998). Moreover, other recent studies in the Southern Ocean have highlighted that diatom community composition could better explain differences in silicification than physiological response to iron enrichment (see Baines et al., 2010; Assmy et al., 2013).

**Various minor comments:**

- UCDW/LCDW identification (L3-5, p5): It is UCDW that is associated with the O2 minimum, not LCDW (e.g. Talley 2013, Oceanography 26, 80-97)

We will correct that in the text.

- Sections 2.2 and 2.3 both have the same heading. These should be "Sample collection" and "Sample preparation" respectively. The sub-sections of Section 2.3 are numbered incorrectly. On L6, p6, could there be an error in the units given for NaOH concentration?

Yes, we will modify as: "2.2 Sample collection" and "2.3 Sample preparation" and change the numbers in section 2.3. The unit given for NaOH concentration is 0.2 mol L-1 and not  $\mu$ mol L-1 as written in the text and will be corrected.

- Section 2.5: it would be better to refer to "full external" rather than "global" reproducibility.

*Ok, we will change "global reproducibility" by "full external reproducibility".*

- Section 3.1: Cao et al. 2012 is not a Southern Ocean study.

We agree, but they have made a figure (Fig.6) that compiled all the  $\delta^{30}$ Si ranges of values for the dissolved and particulate pools in the global ocean (included in the Southern Ocean). We will more clearly mention this reference for their compilation.

- Section 3.3 and Fig. 3: The comparison with late summer conditions during KEOPS-1 is useful, but Fig. 3 is confusing since it combines the seasonal signal (i.e. KEOPS-1 vs. KEOPS-2) with a regional/biogeochemical signal (i.e. HNLC versus Plateau). The figure should separate these regions into separate panels and also show concentration profiles to provide context for the isotope data.

As previously suggested, we will add concentrations profiles in Fig.3. But we will probably not separate the regions in different panels as we have used different colors to discriminate between the HNLC and Plateau profiles.

- Section 3.3.4: It would be good to mention once again that the sediment trap is located at Station A3. On L33, p16, I don't understand why the ML concentration estimated from sediment trap data is compared with mid-November surface concentrations at station E5, rather than with mid-November concentrations at the station where the sediment trap is located (A3-2). Concentrations here are about 2x higher than estimated from the sediment trap  $\delta$ 30SiBSi data. Also, on L29, p17, it should be mentioned that the high values seen in late summer cannot be explained by the steady-state model.

Yes, we will change the sentence P16 L33 by "The remaining DSi concentration in the ML would be 9.53  $\mu$ mol L-1 in November 2011 which is half of the ML DSi concentration measured there mid-November but which is close to the 12  $\mu$ mol L-1 measured at E5." We will also specify that the high values seen in late summer cannot be explained by the steady state model.

- Finally, although in general the manuscript's English is good, it would profit from being proof-read by a native English speaker in order to iron out small grammatical errors.

*Ok, we will take into consideration this comment.*

Moreover, even though none of the reviewers has raised concerns on the accuracy of the data, we would like to add a section in the revised version of the supplementary material. Indeed, there have been some concerns regarding Si isotopic offsets among labs on North Atlantic data (see Brzezinski and Jones, 2015) as well as more recently on the GEOTRACES intercalibration (Grasse et al., to be submitted). We will show in the revised version that we have compared the deep samples of our KEOPS-2 study using a Neptune+ MC-ICP-MS (Closset et al. 2015) and a chemical purification adapted from Hughes et al. (2011): cationic exchange + anion doping with data from KEOPS-1 (Fripiat et al. 2011) using a Nu Instrument MC-ICP-MS (Cardinal et al. 2003) and chemical purification adapted from De La Rocha et al. (1996). On a  $\delta^{30}$ Si vs. 1/Si plot the two dataset show an offset of only 0.08 ‰ which is not analytically significant. We propose to modify the Supplementary Material Figure S4 to better highlight this comparison.

**List of relevant changes in the revised manuscript**

\* P7-8: We have removed the paragraph "KEOPS-2, cruise and hydrological settings" in the Material and Methods and we have combined it with "General considerations" in the Results and Discussion. We have detailed the different groups of stations in this paragraph.

\* We have added a table with station settings in the revised manuscript (Table 1).

\* P8: We have included a paragraph about the accuracy of our data and their comparison with previous  $\delta^{30}$ Si values in this region, and we have modified the corresponding figure (Fig. S4).

\* P11: As suggested by the reviewers, we have changed our interpretation of low  $\delta^{30}$ SiBSi and in surface compared to the subsurface. We have included a comparison with the meridional overturning circulation.

\* P12: we have detailed the reason why we used preferentially the steady-state model to calculate BSi production in the Kerguelen region.

\* P14: We have included the interpretation suggested by a reviewer about isopycnal mixings for explaining the winter water DSi and  $\delta^{30}$ Si signatures.

\* P15: We have modified the paragraph about the decoupling between the dissolved and the particulate phase to make it clearer.

\* P17-18: As suggested by the reviewers, we have detailed the assumptions behind our estimations of BSi production and DSi consumption.

\* P20: We have added some references to support our conclusions>

\* Figures: We have simplified figure 1 and corrected figure 2. We have added 2 panels on figure 3 and included the uncertainties on figure 4.

Marked-up manuscript version (changes are in green color)

[revised manuscript text omitted]